# ZooPFL: Exploring Black-box Foundation Models for Personalized Federated Learning

## Abstract

When personalized federated learning (FL) meets large foundation models, new challenges arise from various limitations in resources. In addition to typical limitations such as data, computation, and communication costs, access to the models is also often limited. This paper endeavors to solve both the challenges of *limited resources* and *personalization*. i.e., distribution shifts between clients. To do so, we propose a method named **ZooPFL** that uses **Z**eroth-**O**rder **O**ptimization for **P**ersonalized **F**ederated **L**earning. ZooPFL avoids direct interference with the foundation models and instead learns to adapt its inputs through zeroth-order optimization. In addition, we employ simple yet effective linear projections to remap its predictions for personalization. To reduce the computation costs and enhance personalization, we propose input surgery to incorporate an auto-encoder with low-dimensional and client-specific embeddings. We provide theoretical support for ZooPFL to analyze its convergence. Extensive empirical experiments on computer vision and natural language processing tasks using popular foundation models demonstrate its effectiveness for FL on black-box foundation models.

## 1 Introduction

In recent years, the growing emphasis on data privacy and security has led to the emergence of federated learning (FL) (Warnat-Herresthal et al., 2021; Chen & Chao, 2022; Chen et al., 2023b; Castiglia et al., 2023; Rodríguez-Barroso et al., 2023; Kuang et al., 2023). FL enables collaborative learning while safeguarding data privacy and security across distributed clients (Yang et al., 2019). However, FL faces two key challenges: *limited resources* and *distribution shifts* (Figure 1 (a, b)).

The rise of large foundation models (Bommasani et al., 2021) has amplified these challenges. The computational demands and communication costs associated with such models hinder the deployment of existing FL approaches (Figure 1a). [1] Most of them require fine-tuning the models on every client.[2] Moreover, foundation models, often proprietary (Van Dis et al., 2023; Sun et al., 2022), grant only *black-box* access, making FL resource-efficient applications a pressing research area.

Recent efforts in FL (Xu et al., 2023b; Zhao et al., 2023; Chen et al., 2023d; Li et al., 2023) have attempted to reduce the number of optimized parameters to minimize computational and communication costs. As illustrated in Figure 1 (c), existing methods use prompts (Liu et al., 2023) or adapters (Cai et al., 2022) to fine-tune foundation models (Xu et al., 2023b). Other approaches (Yurochkin et al., 2019; Liu et al., 2022) focus on limiting the number of communication rounds. All of them however depend on white-box access to the foundation models. On the other hand, distribution shifts are an additional challenge for FL since the data across clients is not necessarily i.i.d. (Li et al., 2020b; Vemulapalli et al., 2023) (Figure 1b). Directly aggregating information e.g., with FedAVG (McMahan et al., 2017) often results in slow convergence and poor performance in each client (Gao et al., 2022). Some methods have been designed to address the personalization of large foundation models (Li et al., 2021; Setayesh et al., 2023; Xu et al., 2023a). However, they

---

[1]Communication costs can be estimated as $C = p \times K \times T$, where $p, T, K$ respectively denote the number of parameters, communication rounds, and clients. With GPT-3 for example (Brown et al., 2020), $p = 175$ billion parameters, making the communication of entire models impractical.

[2]For example, training GPT-2-small (Radford et al., 2019) requires at least two A100 GPUs for 16 hours, a resource unavailable to many.

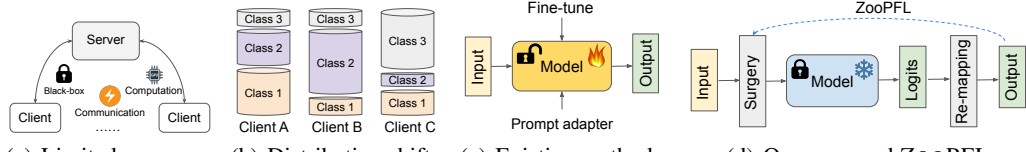

(a) Limited resources  (b) Distribution shifts  (c) Existing methods  (d) Our proposed ZOOPFL

Figure 1: ZOOPFL addresses federated learning with foundation models while coping with limited resources in communication, computation, and model accessibility (a) and being robust to distribution shifts (b). Most existing methods rely on white-box model access (c). In contrast, ZOOPFL is applicable to black-box models by using input surgery and semantic output re-mapping (d).

cannot deal with black-box models. The method proposed in this paper is designed to cope with label shift, i.e. variations in the distribution of labels among clients (Figure 1b).

In this paper, we propose ZOOPFL to cope with limited resources and personalization for federated learning and black-box foundation models. To cope with black-box models, ZOOPFL proposes two strategies, *input surgery* and *semantic re-mapping*, and learning through zeroth-order optimization (ZOO). To reduce the computational costs of ZOO and share information among clients, we employ an auto-encoder with low embedding dimensions to represent transformations. For better personalization, the client-specific embeddings and semantic re-mapping are preserved by each client. Figure 1 (d) illustrates that our proposed method learns transformations on the inputs and mappings of the outputs through *zeroth-order optimization* (Liu et al., 2020; Wang et al., 2018; Lian et al., 2016). This bears similarities with model reprogramming (Chen, 2022) and Reprogrammable-FL (Arif et al., 2023), but the latter is unsuitable for black-box models and personalization. To the best of our knowledge, our method is the first to achieve federated learning with large black-box models, a challenging setting that is becoming increasingly relevant to the real world.

In summary, our contributions are four-fold.

1. **Scenario Exploration**: We delve into the challenges posed by fully black-box foundation models in FL. Our contribution lies in understanding and navigating this complex scenario.
2. **ZOOPFL Framework**: We introduce ZOOPFL, a comprehensive solution tailored for FL in resource-constrained and personalized settings. This framework encompasses **Input Surgery** and **Semantic Re-mapping**. ZOOPFL employs strategic input manipulations, leveraging dedicated embeddings, and employing zeroth-order optimization while it project outputs for specific task and personalization.
3. **Theoretical Support**: We provide formal theoretical support, enhancing ZOOPFL's credibility and offering insights into its workings.
4. **Empirical Validation**: ZOOPFL is rigorously evaluated through computer vision and natural language processing experiments, demonstrating the effectiveness and versatility of ZOOPFL.

## 2 RELATED WORK

**Federated learning** makes it possible to perform distributed multi-party computing without comprising privacy (Zhang et al., 2021; Voigt & Von dem Bussche, 2017; McMahan et al., 2017; Yang et al., 2019; Tariq et al., 2023; Wang et al., 2023a; So et al., 2023). When meeting non-iid data, common FL methods, e.g. FedAVG (McMahan et al., 2017) can suffer from low converge speed and terrible **personalization** performance (Sattler et al., 2019). Specific methods, e.g. FedProx (Li et al., 2020b) and FedBN (Li et al., 2021), are proposed for personalization while additional methods, e.g. (Chen & Chao, 2022; Gupta et al., 2022; Qu et al., 2022), also consideration generalization.

The above methods can fail when entering the era of **large foundation models** (Bommasani et al., 2021; Xing et al., 2023; Zhuang et al., 2023), due to novel issues, e.g. limited resources that make operations on the whole network impossible (Chen et al., 2022; 2023a; Ding et al., 2023). Recent work (e.g. FedPrompt (Zhao et al., 2023),PromptFL (Guo et al., 2023), pFedPG (Yang et al., 2023a), FwdLLM (Xu et al., 2023b)) were proposed to tune part of the whole network for efficiency. However, they all require access to foundation models, which can be impossible in reality.

Table 1: Comparisons of different methods.

| Type | Method | Model scale | Model accessibility | Communication | Computation | Personalization |
|---|---|---|---|---|---|---|
| Base | FedAVG FedBN | Limited | White-box | Inefficient | High | Unsupported Supported |
| Large model for FL | FedPrompt, FedCLIP PromptFL, pFedPG | Unlimited | White-box | Efficient | Low High | Supported |
| Zero-order for FL | FwdLLM, BAFFLE FedZO | Unlimited Limited | White-box | Efficient | Low | Supported |
| Model reprogramming | Reprogrammable-FL | Unlimited | White-box | Efficient | High | Supported |
| Black-box foundation FL | ZooPFL (Ours) | Unlimited | Black-box | Efficient | Low | Supported |

In addition to data privacy, **model privacy** also raises attention (Mo et al., 2020), which means foundation models can be black-box models (Guidotti et al., 2018; Ljung, 2001). Little work paid attention to finetuning or optimizing in this field, but most related work focused on attacks (Yang et al., 2023b;c). One related work is FedZO (Fang et al., 2022) which utilized **zero-order optimization** (Ghadimi & Lan, 2013), but it did not consider utilizing large foundation models.

**Model reprogramming** (MR) (Tsai et al., 2020; Xu et al., 2023c; Chen, 2022) provides a similar solution to ZooPFL and it also focuses on coping with inputs and outputs. Reprogrammable-FL (Arif et al., 2023) adapted MR to the setting of differentially private federated learning. But it preserved local input transformations and shared output transformation layers, which were totally in contrast to ours. Table 1 provides a comprehensive comparison between existing methods and ours. For more detailed related work, please refer to section D.

## 3 METHODOLOGY

In this section, we articulate our proposed ZooPFL. We begin with problem formulation in Sec. 3.1. Then, we show the motivation of designing ZooPFL in Sec. 3.2. Next, Sec. 3.3 introduces the details of our approach. Finally, we propose some discussions in section 3.4.

### 3.1 PROBLEM FORMULATION

We assume there are $n$ different clients $\{C_1, \cdots, C_n\}$ in personalized federated learning scenarios. Each client $C_i$ has its own data $\mathcal{D}_i = \{\mathbf{x}_{i,j}, y_{i,j}\}_{j=1}^{n_i}$ where $n_i$ means the number of data in the $i$th client. Data in different clients have different distributions, i.e. $P(\mathcal{D}_i) \neq P(\mathcal{D}_j)$. In the personalized FL setting, there exists the same black-box large foundation model in each client, $g$, which we know nothing inside and can only obtain logit outputs with fixed-size inputs. Our goal is to achieve personalized (i.e., satisfying) performance with black-box foundation models on each client by learning a significant transformation $s_i$ on inputs and a re-mapping $r_i$ on outputs without accessing $g$ for each client $\mathcal{D}_i$. Specifically, denote $\ell$ a loss function, the learning objective is:

$$\min_{s_i, r_i} \frac{1}{n} \sum_{i=1}^{n} \frac{1}{n_i} \sum_{j=1}^{n_i} \ell(r_i(g(s_i(\mathbf{x}_{i,j}))), y_{i,j}). \tag{1}$$

### 3.2 INTUITION

**Input designs affect the performance of foundation models.** Different representations with the same inputs can induce foundation models to make completely different predictions, which illustrates that adding interference or reconstructing inputs can be utilized for adaptation. However, most methods that add interference are performed at the sample level (White et al., 2023; Cao et al., 2023; Liu & Chilton, 2022; Arif et al., 2023; Zhou et al., 2022; Gal et al., 2022), i.e. special design for each sample, that are unsuitable to exchange information among clients and cannot cope with unseen samples. Therefore, it is necessary to reconstruct samples via an auto-encoder to adapt input with unchanged dimensions for foundation models. The exchange of auto-encoder parameters can facilitate the sharing of input transformation information across different clients.

**Semantic re-mapping generates more semantically meaningful logits.** Although large foundation models have been trained on a huge amount of samples (Radford et al., 2021), there still exists

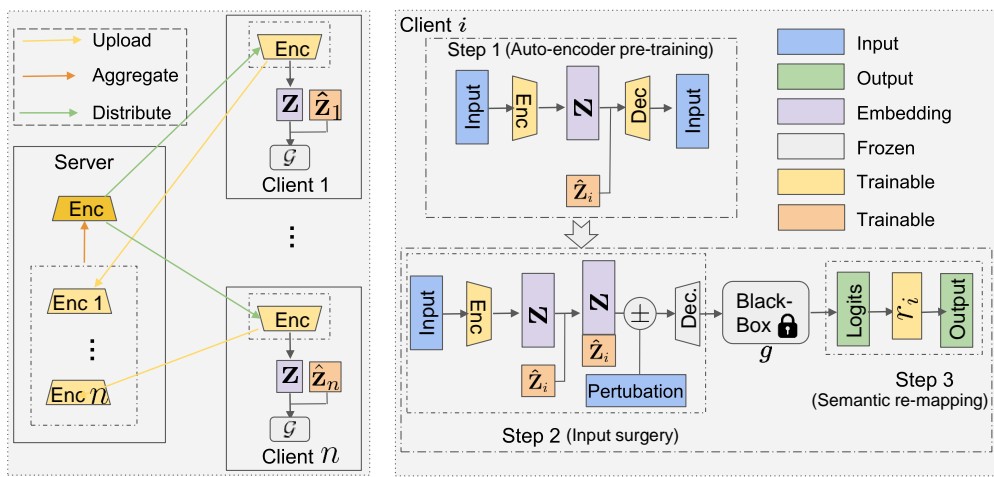

(a) Communication of server-client      (b) Training process in client $i$

Figure 2: The framework of ZooPFL. Please note that communications occur during step 2.

some classes or situations that foundation models cannot cover (Wang et al., 2023b). However, these new scenarios or classes can be made of existing fundamental elements or similar to some existing categories, which means foundation models can be able to extract remarkable features.[3] Considering layers between remarkable features and final logits as random projecting, re-mapping outputs with a simple linear layer can achieve acceptable performance similar to Huang et al. (2015).

**Design logic.** Since access to foundation models is restricted, we have to rely on zeroth-order optimization methods to train auto-encoders, which leads that directly operating on the outputs of auto-encoders with high dimensions can exhaust unaffordable computational costs. To reduce the costs, we fix decoders and compute differences on embedding with low dimensions. For better personalization, we preserve semantic re-mapping in clients. Specifically, we preserve a client-specific embedding, i.e., a simple one-dimensional vector, for each client, which can be concatenated with embedding to generate adapted inputs with personalized characteristics.

### 3.3 ZooPFL

In this paper, we propose ZooPFL to learn input surgery and semantic re-mapping for black-box large foundation models in federated learning. ZooPFL aims to adapt inputs to models and project outputs to meaningful semantic space. ZooPFL mainly consists of three steps, namely, auto-encoder pre-training, input surgery, and semantic re-mapping.[4] Figure 2 shows the pipeline of our approach, where Figure 2(a) describes the communications between clients and the server and Figure 2(b) provides details on how to perform training on a local client. The algorithm flow is in section E.1. Besides, more practical insights can be found in section A.

The training process on a client is described as follows, where steps 2∼3 are iterative.

1. Auto-encoder pre-training: this step directly utilizes inputs to pre-train the auto-encoder which then serves as the input surgery function.
2. Input surgery: this step only updates the encoder of auto-encoder and client-specific embeddings to transform the input consistent with the foundation model.
3. Semantic re-mapping: this step endeavors to re-map logits into meaningful semantic spaces with a simple linear projection.

---

[3]Some popular language models such as BERT (Devlin et al., 2018) and GPT-2 (Radford et al., 2019) in Huggingface utilize a random projection between extracted features and logits.

[4]Note that the pre-training here is different from the pre-training of large foundation models such as self-supervised pre-training. This step is much more efficient than pre-training a large foundation model since we only train an auto-encoder with few layers.

**Auto-encoder Pre-training.** Before input surgery and semantic re-mapping that are assisted by labels, ZOOPFL firstly utilizes inputs of samples to pre-train auto-encoders for better initial understanding of client data and we will fix decoders in the next two steps. For client $C_i$, we denote $\hat{\mathbf{z}}_i$ as the $i$th client-specific embedding and $s_i = o_i \circ q_i$ where $q_i$ and $o_i$ represent the encoder and the decoder respectively. This step is unsupervised and each client utilizes MSE loss to train local $s_i$:

$$\ell_{MSE} = \mathbb{E}_{(\mathbf{x},y)\sim\mathbb{P}(\mathcal{D}_i)} \left\| o_i([q_i(\mathbf{x}), \hat{\mathbf{z}}_i]) - \mathbf{x} \right\|_2^2, \tag{2}$$

where $[\cdot, \cdot]$ denotes the concatenation operation. The updated encoder and decoder of each client are then transmitted to the server. Similar to FedAVG (McMahan et al., 2017), the server aggregates the collected auto-encoders and distributes the aggregated one, $s$, to each client.

$$w(s) = \frac{1}{n} \sum_{i=1}^{n} w(s_i), \tag{3}$$

where $w(s)$ represent parameters of $s$. We assume that all clients contribute equally and participate in training. The above pre-training is iterative and we can obtain well-trained auto-encoders finally.

**Input Surgery.** After pre-training, input surgery optimizes encoders, $q_i$, to transform inputs consistent with foundation models. This step only exchanges encoders of clients to share common knowledge while each client preserves a client-specific embedding to represent personalized knowledge. As shown in Figure 2, the foundation model, $g$, is black-box and the decoder is frozen. In the following, we elaborate on the whole training process in local clients.

In client $C_i$, an input $\mathbf{x}$ is first fed into the encoder $q_i$, generating an embedding vector $\mathbf{z} = q_i(\mathbf{x})$. Then we concatenate $\mathbf{z}$ with the client-specific embedding, $\hat{\mathbf{z}}_i$, and obtain the final embedding feature, $\tilde{\mathbf{z}} = [\mathbf{z}, \hat{\mathbf{z}}_i]$, which is then sent to the decoder. Once processed by the decoder, we can obtain $\tilde{\mathbf{x}} = o_i(\tilde{\mathbf{z}})$ with the same dimension as $\mathbf{x}$, and then the adapted input, $\tilde{\mathbf{x}}$, goes through the foundation model and the re-mapping layer, which generates the final prediction, $\tilde{\mathbf{y}}$. We utilize the cross-entropy loss $\ell_{cls}$ to guide the optimization:

$$\ell_1 = \mathbb{E}_{(\mathbf{x},y)\sim\mathbb{P}(\mathcal{D}_i)} \ell_{cls}(r_i(g(o_i([q_i(\mathbf{x}), \hat{\mathbf{z}}_i]))), y). \tag{4}$$

However, the above objective cannot be directly optimized using the standard stochastic gradient descent since the foundation model $g$ is frozen, preventing us from computing its gradient using back-propagation. We adopt the zeroth-order optimization method, specifically, the coordinate-wise gradient estimate (CGE), to learn $q_i$ and $\hat{\mathbf{z}}_i$ (Zhang et al., 2022; Tu et al., 2019; Liu et al., 2018; Lian et al., 2016; Ghadimi & Lan, 2013). To make the process clear and easy to understand, we freeze $r_i$ and view $o_i$, $g$, and $r_i$ as a whole module, $\mathcal{G}$, in this step.

Assume $\mathbf{z} \in \mathbb{R}^{d_1}$ and $\hat{\mathbf{z}}_i \in \mathbb{R}^{d_2}$. According to CGE, by adding a perturbation to $\tilde{\mathbf{z}}$, we obtain the new embedding and the corresponding classification loss,

$$\tilde{\mathbf{z}}_1 = \tilde{\mathbf{z}} + \rho \mathbf{e}_j, \ell_{\mathbf{x},1} = \ell_{cls}(r_i(g(o_i(\tilde{\mathbf{z}}_1, y)))), \tag{5}$$

where $\mathbf{e}_j = (0, 0, \cdots, 1, 0, 0 \cdots, 0) \in \mathbb{R}^{d_1+d_2}$ denotes the $j$th elementary basic vector and $\rho$ is a hyperparameter that describes the extent of the perturbation. Similarly, we can obtain $\tilde{\mathbf{z}}_2$ and $\ell_{\mathbf{x},2}$.

$$\tilde{\mathbf{z}}_2 = \tilde{\mathbf{z}} - \rho \mathbf{e}_j, \ell_{\mathbf{x},2} = \ell_{cls}(r_i(g(o_i(\tilde{\mathbf{z}}_2, y)))). \tag{6}$$

Then, we have the gradient of $\mathcal{G}$ w.r.t. $\tilde{\mathbf{z}}$ computed as:

$$\nabla_{\tilde{\mathbf{z}}}\mathcal{G}(\tilde{\mathbf{z}}) = (\nabla_{\tilde{\mathbf{z}}}\mathcal{G}(\tilde{\mathbf{z}})_1, \nabla_{\tilde{\mathbf{z}}}\mathcal{G}(\tilde{\mathbf{z}})_2) \approx \sum_{i=1}^{d_1+d_2} \frac{\ell_{\mathbf{x},2} - \ell_{\mathbf{x},1}}{2 \times \rho} \mathbf{e}_j. \tag{7}$$

For $\hat{\mathbf{z}}_i$, we directly update it with corresponding parts of $\nabla_{\tilde{\mathbf{z}}}\mathcal{G}(\tilde{\mathbf{z}})$ via a learning rate $\gamma_2$, $\hat{\mathbf{z}}_i^{new} = \hat{\mathbf{z}}_i - \gamma_2 \times \nabla_{\tilde{\mathbf{z}}}\mathcal{G}(\tilde{\mathbf{z}})_2$ where $\nabla_{\tilde{\mathbf{z}}}\mathcal{G}(\tilde{\mathbf{z}})_2$ denotes the last $d_2$ dimensions of $\nabla_{\tilde{\mathbf{z}}}\mathcal{G}(\tilde{\mathbf{z}})$. For $\nabla_{\tilde{\mathbf{z}}}\mathcal{G}(\tilde{\mathbf{z}})_1$, we can update $q_i$ with the chain rule for differentiation.

$$\nabla_{q_i}\ell_1 = \frac{d\tilde{\mathbf{z}}}{dq_i} \frac{d\mathcal{G}}{d\tilde{\mathbf{z}}} \approx \frac{d\mathbf{z}}{dq_i} \nabla_{\tilde{\mathbf{z}}}\mathcal{G}(\tilde{\mathbf{z}})_1 \approx \frac{d\nabla_{\tilde{\mathbf{z}}}\mathcal{G}(\tilde{\mathbf{z}})_1\mathbf{z}}{dq_i}. \tag{8}$$

Finally, we can update the encoder, $w(q_i^{new}) = w(q_i) - \gamma_1 \times \nabla_{q_i}\ell_1$. Once all clients have updated encoders, we can aggregate encoders in the server and then distribute the aggregated encoder:

$$w(q) = \frac{1}{n} \sum_{i=1}^{n} w(q_i^{new}). \tag{9}$$

**Semantic Re-mapping.** In the last step, we train the encoder that enables the input consistent with foundation models. Here, we perform semantic re-mapping similar to Huang et al. (2015). This step only occurs in each client and no communication exists for simplicity and personalization. We view all parts before $r_i$ as a whole module, $\mathcal{F}$, with two functions, including extracting features and mapping extracted features to a random space and we freeze $\mathcal{F}$. These two functions correspond to artificial features and the first layer in (Huang et al., 2015) respectively and we only update $r_i$ corresponding to the second layer of ELM:

$$\ell_2 = \mathbb{E}_{(\mathbf{x},y)\sim\mathbb{P}(\mathcal{D}_i)}\ell_{cls}(r_i(\mathcal{F}(\mathbf{x})), y). \tag{10}$$

Since this part is behind $g$, $w(r_i)$ can be updated directly.

### 3.4 DISCUSSION

We perform step 2 and step 3 iteratively. There also exist other zeroth-order optimization methods, e.g., the randomized gradient estimate (RGE) (Liu et al., 2020). However, the concrete implementation of zeroth-order optimization is not our focus and we thereby choose CGE for deterministic and stability (Liu et al., 2018). In this paper, we assume large foundation models exist on clients in the form of encrypted assets and we do not need to upload transformed inputs. Moreover, we do care about communication costs and GPU demands instead of training time in each client. To reduce training time in each client, some techniques, such as RGE, random selections on $\mathbf{e}_j$, reduction on $d_1 + d_2$, etc., can be adopted and we leave this as our future work. Our algorithm converges and the asymptotic convergence rate is $\mathbf{O}(\frac{1}{\sqrt{T}})$. For detailed theoretical analysis and the corresponding proofs, please refer to section B and section C.

## 4 EXPERIMENTS

### 4.1 SETUP

**Datasets and baselines.** We evaluate ZOOPFL on 8 popular classification benchmarks with two modalities including computer vision (CV) and natural language processing (NLP). The benchmarks are COVID-19 (Sait et al., 2020), APTOS (Karthik, 2019), Terra100 (Beery et al., 2018), Terra46 (Beery et al., 2018), SST-2 (Wang et al., 2019; Socher et al., 2013), COLA (Wang et al., 2019; Warstadt et al.,

Table 2: Information of benchmarks.

| Modality | Dataset | Samples | Classes | Clients | Selected Samples |
|---|---|---|---|---|---|
| CV | COVID-19 | 9,198 | 4 | 20 | 9,198 |
| | APTOS | 3,662 | 5 | 20 | 1,658 |
| | Terra100 | 5,883 | 10 | 20 | 5,883 |
| | Terra46 | 4,741 | 10 | 20 | 4,741 |
| NLP | SST-2 | 67k | 2 | 20 | 9,763 |
| | COLA | 8.5k | 2 | 20 | 5,700 |
| | Finanical | 4,840 | 3 | 10 | 3,379 |
| | Flipkart | 205,053 | 3 | 20 | 3,048 |

2019), Financial-phrasebank (Financial) (Malo et al., 2014), and Flipkart (Vaghani & Thummar, 2023). Brief information can be found in Table 2.[5] We filter meaningless samples and select samples for global class balance. The concrete select strategies and more details can be found in section F.1 while data distributions can be found in section F.2. To our best knowledge, no other methods are proposed and thereby we only compare our methods with zero-shot pre-trained models (ZS).

**Implementation details.** For vision tasks, we set $g$ as CLIP (Radford et al., 2021) with ResNet50 as the image backbone (Radford et al., 2021). $r$ is a linear layer with dimension $M$ where $M$ is the number of classes. $q$ contains several blocks composed of a convolution layer, a RELU activation layer, a Batch Normalization layer, and a Pooling layer while $o$ contains several blocks composed of a convTranspose layer, a RELU activation layer, and a Batch Normalization layer. We set $d_1 = 6 \times 7 \times 7 = 294$ and $d_2 = 2 \times 7 \times 7 = 98$. We set the learning rate for pretraining as $10^{-4}$ and set other learning rates as hyperparameters. For simplicity, other learning rates are all the same. We set the local epoch number as 1 and set the global round number $T = 120$. Moreover, we do not tune $\rho$ but set $\rho = 5 \times 10^{-3}$. We select the best results according to accuracy on validation parts.

For language tasks, we select four foundation models, including ALBERT-base (Lan et al., 2020), BERT-base (Devlin et al., 2018), DeBERTa-base (He et al., 2021), and GPT2 (Radford et al., 2019). Note that there are recent large language foundation models such as Llama (Touvron et al., 2023)

---

[5]We have chosen so many clients because it reflects the typical real-world scenario where there are numerous clients, each with relatively small amounts of data (Xu et al., 2023b).

and Falcon (Penedo et al., 2023), but we can only experiment with the above ones due to constrained computational devices.[6] Our method works for all kinds of foundation models in various sizes. $q$ simply contains several linear layers followed by batch normalization layers. Please note that we transform input embeddings processed by foundation models for NLP instead of original texts. We set $d_1 = 128 - 32 = 96$ and $d_2 = 32$. We set the local epoch number as 1 and set the global round number $T = 130$. Other settings are similar to computer vision. For concrete structures of auto-encoders, please refer to section F.3.

## 4.2 EXPERIMENTAL RESULTS

Figure 3 shows the results on all eight benchmarks and detailed results are in section F.4. From these results, we have the following observations. 1) Our method achieves the best results on average for all benchmarks whatever the backbone is. It significantly outperforms the zero-shot method with remarkable improvements. In computer vision benchmarks, the improvements are about $42\%$, $25\%$, $21\%$, and $59\%$ for COVID-19, Terra100, APTOS, and Terra46 respectively. In natural language processing benchmarks, for SST-2, COLA, and Flipkart, Financial-phrasebank, the improvements are about $43\%$, $39\%$, $15\%$, and $33\%$ respectively. Please note that there only exist a few training data in each client (for COVID-19, each client only has about 50 samples.), which means utilizing foundation models is important. 2) Our method achieves the best accuracy in most clients, demonstrating the necessity of input surgery and semantic re-mapping. As shown in Figure 3(a)-(d), ZOOPFL only performs slightly worse than ZS in few clients, e.g. client 13 on COVID-19, which can be due to the instability of zeroth-order optimization. 3) For natural language processing, different backbones bring different performance. From Figure 3(g), we can see that our method based on GPT2 can achieve better results compared to other backbones, although ZS performs the worst with GPT2. However, from Figure 3(f), we can see that ZOOPFL based on GPT2 does not achieve the best performance. 4) Why large foundation models cannot achieve acceptable performances on these benchmarks? For computer vision, we choose COVID-19, APTOS, and Terra Incognita and these datasets can be missing during pretraining of CLIP, which leads the failure of CLIP with zero-shot. For natural language processing, although large foundation models can extract remarkable features, they need to be fine-tuned for downstream tasks, which means they may randomly guess without the post-processing. Due to these factors, post-processing to large foundation models is necessary, which is just what we explore in this paper.

## 4.3 ANALYSIS AND DISCUSSION

**Ablation Study.** Figure 4(a) and 4(b) give experiments on ablation study and we have following observations. 1) In most situations, each part of our method brings improvements on both CV and NLP. 2) Step 3 is more significant than Step 2. Since Step 3 re-maps outputs, it can offer semantic meanings to foundation models for specific tasks, which is more direct and effective intuitively. Step 2 transforms inputs that still go through foundation models or even random projections, and thereby it is indirect and less effective. However, by combining Step 2 with Step 3, we can achieve further

---

[6]Our hardware is a server with 4 V100 (16G) GPUs, which cannot afford to train larger foundation models.

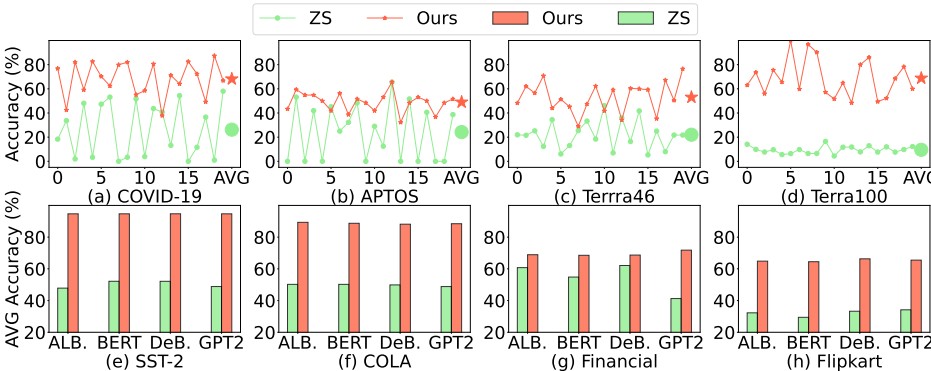

Figure 3: Results on CV (a-d) and NLP (e-h) tasks.

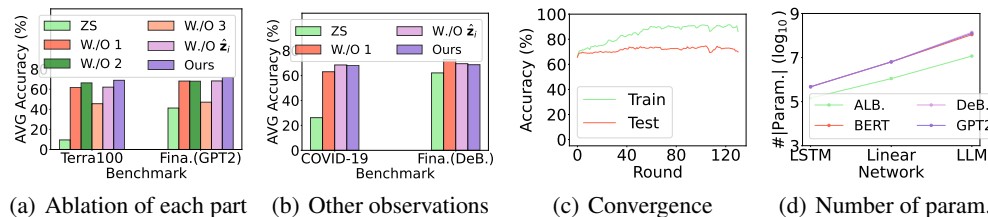

(a) Ablation of each part    (b) Other observations    (c) Convergence    (d) Number of param.

Figure 4: Ablation study, convergence, and communication costs.

Table 3: Resource Consumption.

| Phase | Metric | Foundation Models | ALB. | BERT | DeB. | GPT2 |
|---|---|---|---|---|---|---|
| Inference | FLOPS | Base | 5.4475E+09 | 5.4413E+09 | 7.2483E+09 | 5.0627E+06 |
| | | Ours(Linear) | 5.4497E+09 | 5.4540E+09 | 7.2611E+09 | 1.7772E+07 |
| | | Ours(LSTM) | 5.4656E+09 | 5.4856E+09 | 7.2927E+09 | 4.9377E+07 |
| | Time (S) | Base | 0.4278 | 0.3308 | 0.6475 | 0.6107 |
| | | Ours(Linear) | 0.5856 | 0.4767 | 0.5887 | 0.5321 |
| | | Ours(LSTM) | 0.7582 | 0.6028 | 0.7143 | 0.6753 |
| | Metric | Foundation Models | ALB. | BERT | DeB. | GPT2 |
| Train | GPU Memory (MB) | Base | 5742 | 5380 | 8356 | 7874 |
| | | Ours(Linear) | 1718 | 3086 | 3346 | 3838 |
| | | Ours(LSTM) | 1738 | 2076 | 2336 | 2718 |
| | Metric | Foundation Models | ALB. | BERT | DeB. | GPT2 |
| / | Storage (M) | Base | 45 | 418 | 532 | 475 |
| | | Ours(Linear) | 54 | 467 | 580 | 524 |
| | | Ours(LSTM) | 46 | 421 | 534 | 478 |

improvements. 3) In some situations, client-specific embeddings do not bring remarkable improvements, which can be induced by two reasons. First, CGE is not stable enough and we cannot ensure ZOOPFL finds the best global optimals. Second, to ensure fairness, we offer comparison methods without client-specific embeddings containing larger dimensions and thereby these methods can learn better representations for auto-encoders. 4) Step 1 brings significant improvement for CV while it is less effective for NLP. This can be due to two reasons. We provide better auto-encoders for CV but simple linear layers for NLP. Moreover, the closed pretraining of an auto-encoder without subsequent adjustments to the decoder may not be suitable for NLP. Fortunately, ZOOPFL can achieve convincing improvements compared to ZS no matter whether adopting step 1.

**Convergence and Communication Cost.** We provide convergence analysis and communication cost comparisons in Figure 4(c) and 4(d), respectively. Figure 4(c) shows that both the average training accuracy and testing accuracy are convergent. There exist slight disturbances due to instability of CGE and the process of federated learning. Moreover, we can find that there exists a divergence between training and testing, which means there could be further improvements if more generalization techniques could be adopted which we leave as our future work. From Figure 4(d), we can see exchanging in encoders can reduce a significant amount of transmission cost, especially for LSTM, which means our method can be employed in reality.

**Resource Consumption Analysis** Table 3 shows resource consumption on other metrics. Since our model contains the foundation model on each client, it will slightly consume more FLOPS, inference time (running ten times to calculate the time.), and storage. We can observe that, compared to the foundational model, the incremental changes can be negligible, and sometimes even slightly less resource-intensive (possibly due to the instability of the computing environment). In terms of GPU consumption, although our model is larger, it consumes less resources. It is reasonable that our approach consumes less GPU since we do not need to compute and store the gradients of foundational models.

**More insightful analysis.** 1) **Can stronger backbones bring better performance?** From Figure 5(a) and 4(d), we can see that auto-encoders comprised of LSTM can bring better performance with fewer communications (especially for GPT2), which means more suitable backbones can lead

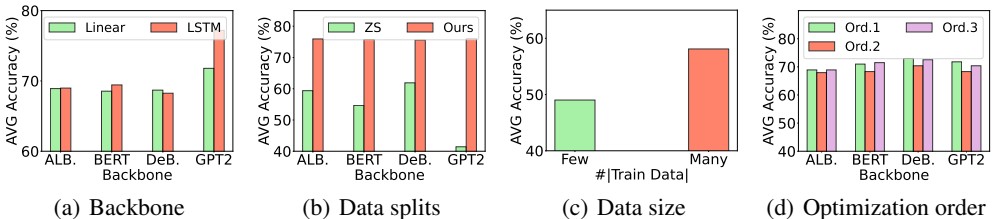

Figure 5: More discussions by varying backbones, data splits, data sizes, and optimization order.

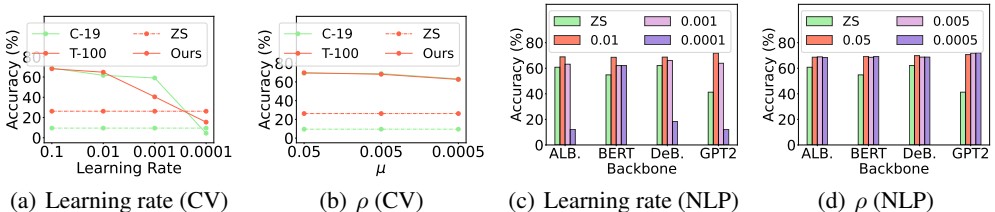

Figure 6: Parameter sensitivity of computer vision and natural language tasks (on finance data).

to better performance. 2) **How can data splits influence the performance?** Figure 5(b) shows that ZOOPFL still achieves better performance when using a different parameter for Dirichlet distribution (0.1 vs. 0.2) for NLP data split. In this more personalized situation, ZS maintains a similar performance while ours performs better. 3) **More training data, better results?** As shown in Figure 5(c), we choose the APTOS dataset where our method has the worst performance, to evaluate the influence of training data. We find that more training data can bring further improvements, which is completely consistent with our intuition. 4) **Can optimization order influence performance?** We provide three orders for optimization. Order 1 is what we adopted. Order 2 is to perform step 2 for all rounds and then perform step 3, which means these two steps are split. In order 3, we first optimize the encoder, then client-specific embeddings, and finally semantic re-mapping layers, and these parts are iterative. As shown in Figure 5(d), Order 1 and Order 3 can perform slightly better than Order 2, which demonstrates the necessity of joint optimization. More experiments on visualizations can be found in section F.5.

**Parameter sensitivity.** Figure 6 provides parameter sensitivity and we obtain following observations. 1) Our method is stable for a wide range of parameters although CGE may lead instability. 2) For most situations, larger learning rates with Adam can bring better performance. 3) ZOOPFL can achieve further improvement if we finetune hyperparameters more carefully. For example, we can choose larger learning rates, e.g.0.5 or choose more suitable $\rho$ for specific tasks, e.g. 0.05 for CV.

## 5 CONCLUSION AND DISCUSSION

We proposed ZOOPFL which can deal with large black-box models in federated learning. ZOOPFL mainly consists of two parts, including input surgery and semantic re-mapping. Moreover, with a client-specific embedding, ZOOPFL can be more personalized. We demonstrated its effectiveness on both CV and NLP tasks. ZOOPFL achieved remarkable performance without large communication costs and high demands of GPUs.

As the first exploration in black-box federated learning for large foundation models, ZOOPFL can be more perfect by pursuing the following avenues. 1) Since the stability and speed of CGE influence the performance of step 2, it can be better to seek more stable and efficient optimization algorithms. 2) Foundation models in ZOOPFL can be enhanced by other ways, e.g., auxiliary models, to serve as a complement to foundation models. 3) Experiments with larger foundation models can be performed for evaluation if computational resources are enough in the future.

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

## A  PRACTICAL INSIGHTS

**What is our setting?**  We view the large foundation models as black-box that can only provide the outputs according to inputs. We have *no access* to any internal information on the foundation models, which means, *no backpropagation* is allowed for updating. Each client preserves the same foundation model *locally*. Most importantly, we do *not consider* the storage of large models and the additional costs associated with inference. We aim to utilize large black-box foundation models for better personalized federated learning.

**Why is it practical?**  To make the best of large foundation models in federated learning, one must:

- either fine-tune or adapt the models in their own data,
- or perform federated learning on the cloud.

Therefore, it is easy to see the value of our work, since:

- **fine-tuning or adapting locally is extremely expensive even if we have many open-source foundation models.** Why? Because fine-tuning on client side requires high computation and communication costs. By "client side", we do not mean just mobile phone devices, but any organization (e.g., a hospital) could be a client to be a part of the FL cycle.
- **federated learning on the cloud is not the ideal solution if you care about privacy.** Why? One cannot trust the cloud providers by uploading all the training data to the cloud. So, the best practice is to perform computation *locally*.

Combining the above situations. i.e., updating models locally with low cost, one can conclude that our proposed black-box FL is the only solution. Specifically, note that "black-box" does not only mean we do not have model access; it is a more broad technique for model update when local BP cannot be performed due to large model sizes.

## B  THEORETICAL ANALYSIS

We present the convergence analysis of ZOOPFL. There exist three parts to optimize during step 2 and step 3, including the parameters of the encoder ($\hat{\mathbf{u}}$), clients specific embeddings ($\mathbf{v}_{1,i}$), and semantic re-mapping layers ($\mathbf{v}_{2,i}$). Following Pillutla et al. (2022), we group parameters into two parts, i.e. $\mathbf{u}_i := \hat{\mathbf{u}} \cup \mathbf{v}_{1,i}$, $\mathbf{v}_i := \mathbf{v}_{2,i}$.[7] Now we give the main conclusion with proofs in section C.

**Theorem 1.** *Suppose assumptions 1, 2, 3, and 4 hold, and the learning rates in* ZOOPFL *are chosen as* $\gamma_{\mathbf{u}} = \eta/(L_{\mathbf{u}}\tau_{\mathbf{u}})$ *and* $\gamma_{\mathbf{v}} = \eta/(L_{\mathbf{v}}\tau_{\mathbf{v}})$, *with*

$$\eta \le \min\left\{\frac{1}{24(1+\mu^2)}, \frac{m}{128\chi^2(n-m)}, \sqrt{\frac{m}{\chi^2 n}}\right\}. \tag{11}$$

*Then, right after the training of $T$ epochs, ignoring absolute constants, we have*

$$\frac{1}{T}\sum_{t=0}^{T-1}\left(\frac{1}{L_{\mathbf{u}}}\mathbb{E}\left[\Delta_{\mathbf{u}}^{(t)}\right] + \frac{m}{nL_{\mathbf{v}}}\mathbb{E}\left[\Delta_{\mathbf{v}}^{(t)}\right]\right) \le \frac{\Delta F_0}{\eta T} + \eta\sigma_{alt,1}^2 + \eta^2\sigma_{alt,2}^2. \tag{12}$$

**Corollary 1.** *An optimal learning rate is:*

$$\eta = \left(\frac{\Delta F_0}{T\sigma_{alt,1}^2}\right)^{1/2} \bigwedge \left(\frac{\Delta F_0^2}{T^2\sigma_{alt,2}^2}\right)^{1/3} \bigwedge \frac{1}{1+\mu^2} \bigwedge \frac{m}{\chi^2(n-m)} \bigwedge \sqrt{\frac{m}{\chi^2 n}}. \tag{13}$$

*We have, ignoring absolute constants,*

$$\frac{1}{T}\sum_{t=0}^{T-1}\left(\frac{1}{L_{\mathbf{u}}}\mathbb{E}\left[\Delta_{\mathbf{u}}^{(t)}\right] + \frac{m}{nL_{\mathbf{v}}}\mathbb{E}\left[\Delta_{\mathbf{v}}^{(t)}\right]\right) \le$$

$$\frac{(\Delta F_0\sigma_{alt,1}^2)^{1/2}}{\sqrt{T}} + \frac{(\Delta F_0^2\sigma_{alt,2}^2)^{1/3}}{T^{2/3}} + \frac{\Delta F_0}{T}\left(1 + \mu^2 + \chi^2(\frac{n}{m}-1) + \sqrt{\chi^2\frac{n}{m}}\right). \tag{14}$$

---

[7]Different from (Pillutla et al., 2022), optimized parameters in ZOOPFL contain three parts and we utilize ZOO instead of gradients.

The measure of convergence of Algorithm 1 is in terms of the weighted average of square norms of the gradients of loss function $\mathbb{E}\left[\Delta_{\mathbf{u}}^{(t)}\right]$ and $\mathbb{E}\left[\Delta_{\mathbf{v}}^{(t)}\right]$ through iterations from 1 to $T-1$, i.e. the left hand of equation 14. As the square norms of the gradients of loss function at the optimal solution is zero, whether or not these norms approach zero is a good criterion of the convergence. With this choice of optimal learning rate, it is clear from the right hand of equation 14 that our algorithm converges and the asymptotic convergence rate is $\mathbf{O}(\frac{1}{\sqrt{T}})$.

## C  PROOFS

The proof is based on the theoretic work of personalized federate learning pioneered in Pillutla et al. (2022). Firstly, we will make some assumptions on our models (parameters) akin to those in Pillutla et al. (2022) with some differences specific to our scenario. One can refer Pillutla et al. (2022) for more details.

Recall that our loss function is defined as follows:[8]

$$F(\hat{\mathbf{u}}, \mathbf{v}) = \frac{1}{n} \sum_{i=1}^{n} F_i(\hat{\mathbf{u}}, \mathbf{v}_i), \tag{15}$$

where $F_i(\hat{\mathbf{u}}, \mathbf{v}_i) = \mathbb{E}_{\mathbf{x} \sim \mathcal{D}_i} f_i(\hat{\mathbf{u}}, \mathbf{v}_i, \mathbf{x})$. $\hat{\mathbf{u}}$ denotes the sharing parameters, i.e. parameters of $s_i$, while $\mathbf{v}_i$ denotes parameters preserved in each client. According to the structure of ZOOPFL, $\mathbf{v}_i$ is also decomposed into two parts: $\mathbf{v}_i = \mathbf{v}_{1,i} \cup \mathbf{v}_{2,i}$ corresponding to parameters of $\hat{\mathbf{z}}_i$ and $r_i$ respectively, and as pointed out in main sections of our paper, we regroup our parameters as follows: for each device $i$, $\mathbf{u}_i := \hat{\mathbf{u}} \cup \mathbf{v}_{1,i}$, $\mathbf{v}_i := \mathbf{v}_{2,i}$.

**Assumption 1.** *(Smoothness). For each device $i = 1, 2, \ldots n$, the object $F_i$ is smooth, i.e., it is continuously differentiable and,*

1. *$\mathbf{u}_i \to \nabla_{\mathbf{u}} F_i(\mathbf{u}_i, \mathbf{v}_i)$ is $L_{\mathbf{u}}$-Lipschitz for all $\mathbf{v}_i$,*

2. *$\mathbf{v}_i \to \nabla_{\mathbf{v}} F_i(\mathbf{u}_i, \mathbf{v}_i)$ is $L_{\mathbf{v}}$-Lipschitz for all $\mathbf{u}_i$,*

3. *$\mathbf{v}_i \to \nabla_{\mathbf{u}} F_i(\mathbf{u}_i, \mathbf{v}_i)$ is $L_{\mathbf{uv}}$-Lipschitz for all $\mathbf{u}_i$, and,*

4. *$\mathbf{u}_i \to \nabla_{\mathbf{v}} F_i(\mathbf{u}_i, \mathbf{v}_i)$ is $L_{\mathbf{vu}}$-Lipschitz for all $\mathbf{v}_i$.*

*Further, we assume for some $\chi > 0$ that*

$$max\{L_{\mathbf{uv}}, L_{\mathbf{vu}}\} \leq \chi \sqrt{L_{\mathbf{u}} L_{\mathbf{v}}}. \tag{16}$$

**Assumption 2.** *(Bounded Gradient). For each device $i = 1, 2, \ldots n$, the object $F_i$ has bounded gradient, that is, there exists $M_{\mathbf{u}}, M_{\mathbf{v}} > 0$ such that*

$$|\nabla_{\mathbf{u}} f_i(\mathbf{u}_i, \mathbf{v}_i, \mathbf{x})| < M_{\mathbf{u}}, \quad and \quad |\nabla_{\mathbf{v}} f_i(\mathbf{u}_i, \mathbf{v}_i, \mathbf{x})| < M_{\mathbf{v}} \quad \forall \mathbf{x} \in \mathcal{D}_i. \tag{17}$$

**Assumption 3.** *(Bounded Variance). Let $\mathcal{D}_i$ denote a probability distribution over the data space $\mathcal{Z}$ on device $i$. There exist functions $G_{i,\mathbf{u}}$ and $G_{i,\mathbf{v}}$ which are unbiased estimates of $\nabla_{\mathbf{u}} F_i$ and $\nabla_{\mathbf{v}} F_i$ respectively. That is, for all $\mathbf{u}_i, \mathbf{v}_i$:*

$$\mathbb{E}_{\mathbf{x} \sim \mathcal{D}_i}[G_{i,\mathbf{u}}(\mathbf{u}_i, \mathbf{v}_i, \mathbf{x})] = \nabla_{\mathbf{u}} F_i(\mathbf{u}_i, \mathbf{v}_i), \quad and \quad \mathbb{E}_{\mathbf{x} \sim \mathcal{D}_i}[G_{i,\mathbf{v}}(\mathbf{u}_i, \mathbf{v}_i, \mathbf{x})] = \nabla_{\mathbf{v}} F_i(\mathbf{u}_i, \mathbf{v}_i). \tag{18}$$

*Furthermore, the variance of these estimators is at most $\sigma_{\mathbf{u}}^2$ and $\sigma_{\mathbf{v}}^2$ respectively. That is,*

$$\mathbb{E}_{\mathbf{x} \sim \mathcal{D}_i} \|G_{i,\mathbf{u}}(\mathbf{u}_i, \mathbf{v}_i, \mathbf{x}) - \nabla_{\mathbf{u}} F_i(\mathbf{u}_i, \mathbf{v}_i)\|^2 \leq \sigma_{\mathbf{u}}^2, \tag{19}$$

$$\mathbb{E}_{\mathbf{x} \sim \mathcal{D}_i} \|G_{i,\mathbf{v}}(\mathbf{u}_i, \mathbf{v}_i, \mathbf{x}) - \nabla_{\mathbf{v}} F_i(\mathbf{u}_i, \mathbf{v}_i)\|^2 \leq \sigma_{\mathbf{v}}^2. \tag{20}$$

In this work, we usually take the particular form $G_{i,\mathbf{u}}(\mathbf{u}_i, \mathbf{v}_i, \mathbf{x}) = \nabla_{\mathbf{u}} f_i((\mathbf{u}_i, \mathbf{v}_i), \mathbf{x})$, which is the gradient of the loss on datapoint $\mathbf{x} \sim \mathcal{D}_i$ under the model $(\mathbf{u}_i, \mathbf{v}_i)$, and similarly for $G_{i,\mathbf{v}}$.

As our model has a black box LLM, we can't get the gradient of parameters in this part. So we resort to zero-order optimization partially. In particular, we take differences of function values

---

[8]As we will regroup parameters later, we use $\hat{\mathbf{u}}$ instead of $\mathbf{u}$ to make symbols consistent.

to estimate unknown gradients in that part. The resulting method is dubbed stochastic difference descent method. More precisely, Let $G$ be a continuous function on $\mathbf{x}$, $\boldsymbol{\rho}$ be a fixed vector with the same dimension of $\mathbf{x}$ and its norm $||\boldsymbol{\rho}|| = \rho$, we denote $\Delta_{\boldsymbol{\rho},\mathbf{x}}G(x) := (G(\mathbf{x} + \boldsymbol{\rho}) - G(\mathbf{x}))/\rho$ or $(G(\mathbf{x} + \boldsymbol{\rho}) + G(\mathbf{x} - \boldsymbol{\rho}) - 2G(\mathbf{x}))/(2\rho)$ to be the difference of $G$ at point $\mathbf{x}$ with step $\boldsymbol{\rho}$. Then the way we update $\mathbf{x}$ is similar to that in stochastic gradient descent method:

$$\mathbf{x}_{k+1} = \mathbf{x}_k - \gamma\Delta_{\boldsymbol{\rho},\mathbf{x}}G(\mathbf{x})|_{\mathbf{x}=\mathbf{x}_k}. \tag{21}$$

Let us denote $\tilde{G}_{i,\mathbf{u}}(\mathbf{u}_i, \mathbf{v}_i, \mathbf{x}) := \Delta_{\boldsymbol{\rho},\mathbf{u}}f_i((\mathbf{u}_i, \mathbf{v}_i), \mathbf{x}), \tilde{G}_{i,\mathbf{v}}(\mathbf{u}_i, \mathbf{v}_i, \mathbf{x}) := \Delta_{\boldsymbol{\rho},\mathbf{v}}f_i((\mathbf{u}_i, \mathbf{v}_i), \mathbf{x})$. Unlike Assumption. 3, $\tilde{G}_{i,\mathbf{u}}(\mathbf{u}_i, \mathbf{v}_i, \mathbf{x})$(resp.$\tilde{G}_{i,\mathbf{v}}(\mathbf{u}_i, \mathbf{v}_i, \mathbf{x})$) is not an unbiased estimation of $\nabla_{\mathbf{u}}F_i(\mathbf{u}_i, \mathbf{v}_i)$ (resp.$\nabla_{\mathbf{v}}F_i(\mathbf{u}_i, \mathbf{v}_i)$). However, under Assumption. 1,Assumption. 2 and Assumption. 3, we have the following estimates:

**Lemma 1.** *(Bounded 1st and 2nd moments).*

$$||\mathbb{E}_{\mathbf{x}\sim\mathcal{D}_i}\tilde{G}_{i,\mathbf{u}}(\mathbf{u}_i, \mathbf{v}_i, \mathbf{x}) - \nabla_{\mathbf{u}}F_i(\mathbf{u}_i, \mathbf{v}_i)|| \leq L_{\mathbf{u}}\rho, \tag{22}$$

$$||\mathbb{E}_{\mathbf{x}\sim\mathcal{D}_i}\tilde{G}_{i,\mathbf{v}}(\mathbf{u}_i, \mathbf{v}_i, \mathbf{x}) - \nabla_{\mathbf{v}}F_i(\mathbf{u}_i, \mathbf{v}_i)|| \leq L_{\mathbf{u}}\rho. \tag{23}$$

*Furthermore,*

$$\mathbb{E}_{\mathbf{x}\sim\mathcal{D}_i}||\tilde{G}_{i,\mathbf{u}}(\mathbf{u}_i, \mathbf{v}_i, \mathbf{x}) - \nabla_{\mathbf{u}}F_i(\mathbf{u}_i, \mathbf{v}_i))||^2 \leq 2L_{\mathbf{u}}^2\rho^2 + 2\sigma_{\mathbf{u}}^2, \tag{24}$$

$$\mathbb{E}_{\mathbf{x}\sim\mathcal{D}_i}||\tilde{G}_{i,\mathbf{v}}(\mathbf{u}_i, \mathbf{v}_i, \mathbf{x}) - \nabla_{\mathbf{v}}F_i(\mathbf{u}_i, \mathbf{v}_i))||^2 \leq 2L_{\mathbf{v}}^2\rho^2 + 2\sigma_{\mathbf{v}}^2. \tag{25}$$

*Proof.* For each $\mathbf{x} \in \mathcal{D}_i$ there exists a $\mathbf{u}_i'$ between $\mathbf{u}_i$ and $\mathbf{u}_i + \boldsymbol{\rho}$ in each component such that $(\nabla_{\mathbf{u}}f_i(\mathbf{u}_i + \boldsymbol{\rho}, \mathbf{v}_i, \mathbf{x}) - \nabla_{\mathbf{u}}f_i(\mathbf{u}_i, \mathbf{v}_i, \mathbf{x}))/\rho = \nabla_{\mathbf{u}}f_i(\mathbf{u}_i', \mathbf{v}_i, \mathbf{x})$. Then by the smoothness assumption:

$$||\nabla_{\mathbf{u}}f_i(\mathbf{u}_i', \mathbf{v}_i, \mathbf{x}) - \nabla_{\mathbf{u}}f_i(\mathbf{u}_i, \mathbf{v}_i, \mathbf{x})|| \leq L_{\mathbf{u}}||\mathbf{u}_i' - \mathbf{u}_i|| \leq L_{\mathbf{u}}||\boldsymbol{\rho}|| = L_{\mathbf{u}}\rho. \tag{26}$$

Thus

$$\tilde{G}_{i,\mathbf{u}}(\mathbf{u}_i, \mathbf{v}_i, \mathbf{x}) - \nabla_{\mathbf{u}}F_i(\mathbf{u}_i, \mathbf{v}_i) = \tag{27}$$

$$\tilde{G}_{i,\mathbf{u}}(\mathbf{u}_i, \mathbf{v}_i, \mathbf{x}) - \nabla_{\mathbf{u}}f_i((\mathbf{u}_i, \mathbf{v}_i), \mathbf{x}) + \nabla_{\mathbf{u}}f_i((\mathbf{u}_i, \mathbf{v}_i), \mathbf{x}) - \nabla_{\mathbf{u}}F_i(\mathbf{u}_i, \mathbf{v}_i). \tag{28}$$

Taking expectation on both sides and applying equation 26 and $\mathbb{E}_{\mathbf{x}\sim\mathcal{D}_i}\nabla_{\mathbf{u}}f_i((\mathbf{u}_i, \mathbf{v}_i), \mathbf{x}) = \nabla_{\mathbf{u}}F_i(\mathbf{u}_i, \mathbf{v}_i)$, we get

$$\mathbb{E}_{\mathbf{x}\sim\mathcal{D}_i}\tilde{G}_{i,\mathbf{u}}(\mathbf{u}_i, \mathbf{v}_i, \mathbf{x}) - \nabla_{\mathbf{u}}F_i(\mathbf{u}_i, \mathbf{v}_i) = \mathbb{E}_{\mathbf{x}\sim\mathcal{D}_i}\tilde{G}_{i,\mathbf{u}}(\mathbf{u}_i, \mathbf{v}_i, \mathbf{x}) - \nabla_{\mathbf{u}}f_i((\mathbf{u}_i, \mathbf{v}_i), \mathbf{x}) \tag{29}$$

$$\leq \mathbb{E}_{\mathbf{x}\sim\mathcal{D}_i}|\tilde{G}_{i,\mathbf{u}}(\mathbf{u}_i, \mathbf{v}_i, \mathbf{x}) - \nabla_{\mathbf{u}}f_i((\mathbf{u}_i, \mathbf{v}_i), \mathbf{x})| \leq L_{\mathbf{u}}\rho. \tag{30}$$

Taking absolute values on both sides completes the proof of equation 22. The same is true for equation 23.

For equation 24 and equation 25, we note that by Cauchy-Schwartz inequality

$$||\tilde{G}_{i,\mathbf{u}}(\mathbf{u}_i, \mathbf{v}_i, \mathbf{x}) - \nabla_{\mathbf{u}}F_i(\mathbf{u}_i, \mathbf{v}_i)||^2 \leq 2||\tilde{G}_{i,\mathbf{u}}(\mathbf{u}_i, \mathbf{v}_i, \mathbf{x}) - \nabla_{\mathbf{u}}f_i(\mathbf{u}_i, \mathbf{v}_i, \mathbf{x})||^2 \tag{31}$$

$$+ 2||\nabla_{\mathbf{u}}f_i(\mathbf{u}_i, \mathbf{v}_i, \mathbf{x}) - \nabla_{\mathbf{u}}F_i(\mathbf{u}_i, \mathbf{v}_i)||^2. \tag{32}$$

Taking expectation on both sides and using equation 26 and Assumption. 3 complete the proof. The same is true for equation 25.

$$\square$$

As our model has the form $F = r \circ J \circ s$ with $s$ corresponding to the encoder, $J$ corresponding to decoder and black-box LLM, and $r$ corresponding to linear remapping, the following corollary is useful.

**Corollary 2.** *Let $s_i, J_i, r_i$ be three continuously differentiable functions such that $f_i(\mathbf{u}_i, \mathbf{v}_i, \mathbf{x}) = r_i \circ J_i \circ s_i$, then we have*

$$||\mathbb{E}_{\mathbf{x}\sim\mathcal{D}_i}\nabla r_{i,\mathbf{u}} \circ \Delta_{\boldsymbol{\rho}}J_{i,\mathbf{u}} \circ \nabla_{\mathbf{u}}s_{i,\mathbf{u}}(\mathbf{u}_i, \mathbf{v}_i, \mathbf{x}) - \nabla_{\mathbf{u}}F_i(\mathbf{u}_i, \mathbf{v}_i)|| \leq M_{\mathbf{u}}^2 L_{\mathbf{u}}\rho, \tag{33}$$

$$||\mathbb{E}_{\mathbf{x}\sim\mathcal{D}_i}\nabla r_{i,\mathbf{u}} \circ \Delta_{\boldsymbol{\rho}}J_{i,\mathbf{u}} \circ \nabla_{\mathbf{v}_i}s_{i,\mathbf{u}}(\mathbf{u}_i, \mathbf{v}_i, \mathbf{x}) - \nabla_{\mathbf{v}}F_i(\mathbf{u}_i, \mathbf{v}_i)|| \leq M_{\mathbf{v}}^2 L_{\mathbf{v}}\rho. \tag{34}$$

*Furthermore,*

$$\mathbb{E}_{\mathbf{x}\sim\mathcal{D}_i}||\nabla r_{i,\mathbf{u}} \circ \Delta_{\boldsymbol{\rho}}J_{i,\mathbf{u}} \circ \nabla_{\mathbf{u}}s_{i,\mathbf{u}}(\mathbf{u}_i, \mathbf{v}_i, \mathbf{x}) - \nabla_{\mathbf{u}}F_i(\mathbf{u}_i, \mathbf{v}_i)||^2 \leq 2M_{\mathbf{u}}^4 L_{\mathbf{u}}^2\rho^2 + 2\sigma_{\mathbf{u}}^2, \tag{35}$$

$$\mathbb{E}_{\mathbf{x}\sim\mathcal{D}_i}||\nabla r_{i,\mathbf{u}} \circ \Delta_{\boldsymbol{\rho}}J_{i,\mathbf{u}} \circ \nabla_{\mathbf{v}_i}s_{i,\mathbf{u}}(\mathbf{u}_i, \mathbf{v}_i, \mathbf{x}) - \nabla_{\mathbf{v}}F_i(\mathbf{u}_i, \mathbf{v}_i)||^2 \leq 2M_{\mathbf{v}}^4 L_{\mathbf{v}}^2\rho^2 + 2\sigma_{\mathbf{v}}^2. \tag{36}$$

*Proof.* Using the bounded gradient assumption and following the steps in the proof of Lemma. 1 completes the proof. □

Finally, we make a gradient diversity assumption.

**Assumption 4.** *(Partial Gradient Diversity). There exists $\delta \geq 0$ and $\mu \geq 0$ such that for all $\mathbf{u}$ and $\mathbf{v}$,*

$$\frac{1}{n} \sum_{i=1}^{n} ||\nabla_{\mathbf{u}} F_i(\mathbf{u}_i, \mathbf{v}_i) - \nabla_{\mathbf{u}} F(\mathbf{u}, \mathbf{v})||^2 \leq \delta^2 + \mu^2 ||\nabla_{\mathbf{u}} F(\mathbf{u}, \mathbf{v})||^2. \tag{37}$$

Please refer Pillutla et al. (2022) for more background on some of these assumptions.

### C.1 CONVERGENCE ANALYSIS OF ZOOPFL

We give the error bounds results of Algorithm 1, thus theoretically establishing the convergence property.

In our case, we rename the parameters so that $\mathbf{u} := \hat{\mathbf{u}} \cup \mathbf{v}_1, \mathbf{v} := \mathbf{v}_2$. As per Appendix A.3 in Pillutla et al. (2022), we use the constants

$$\sigma_{alt,1}^2 = \frac{\delta^2}{L_{\mathbf{u}}} \left(1 - \frac{m}{n}\right) + \frac{2M_{\mathbf{u}}^4 L_{\mathbf{u}}^2 \rho^2 + 2\sigma_{\mathbf{u}}^2}{L_{\mathbf{u}}} + \frac{(2M_{\mathbf{v}}^4 L_{\mathbf{v}}^2 \rho^2 + 2\sigma_{\mathbf{v}}^2)(m + \chi^2(n-m))}{L_{\mathbf{v}} n}, \tag{38}$$

$$\sigma_{alt,2}^2 = \frac{2M_{\mathbf{u}}^4 L_{\mathbf{u}}^2 \rho^2 + 2\sigma_{\mathbf{u}}^2 + \delta^2}{L_{\mathbf{u}}} (1 - \tau_{\mathbf{u}}^{-1}) \tag{39}$$

$$+ \frac{(2M_{\mathbf{v}}^4 L_{\mathbf{v}}^2 \rho^2 + 2\sigma_{\mathbf{v}}^2)m}{L_{\mathbf{v}} n} (1 - \tau_{\mathbf{u}}^{-1}) + \frac{\chi^2(2M_{\mathbf{v}}^4 L_{\mathbf{v}}^2 \rho^2 + 2\sigma_{\mathbf{v}}^2)}{L_{\mathbf{v}}}. \tag{40}$$

and the definitions

$$\Delta_{\mathbf{u}}^{(t)} = \frac{1}{n} \sum_{i=1}^{n} ||\nabla_{\mathbf{u}} F_i \left(\mathbf{u}_i^{(t)}, \mathbf{v}_i^{(t)}\right)||^2, \quad and, \quad \Delta_{\mathbf{v}}^{(t)} = \frac{1}{n} \sum_{i=1}^{n} ||\nabla_{\mathbf{v}} F_i \left(\mathbf{u}_i^{(t)}, \mathbf{v}_i^{(t)}\right)||^2. \tag{41}$$

**Theorem 1.** *Suppose Assumption. 1,Assumption. 2,Assumption. 3 and Assumption. 4 hold and the learning rates in ZOOPFL are chosen as $\gamma_{\mathbf{u}} = \eta/(L_{\mathbf{u}}\tau_{\mathbf{u}})$ and $\gamma_{\mathbf{v}} = \eta/(L_{\mathbf{v}}\tau_{\mathbf{v}})$, with*

$$\eta \leq \min \left\{ \frac{1}{24(1 + \mu^2)}, \frac{m}{128\chi^2(n-m)}, \sqrt{\frac{m}{\chi^2 n}} \right\}. \tag{42}$$

*Then, right after the training of $T$ epochs, ignoring absolute constants, we have*

$$\frac{1}{T} \sum_{t=0}^{T-1} \left( \frac{1}{L_{\mathbf{u}}} \mathbb{E}\left[\Delta_{\mathbf{u}}^{(t)}\right] + \frac{m}{nL_{\mathbf{v}}} \mathbb{E}\left[\Delta_{\mathbf{v}}^{(t)}\right] \right) \leq \frac{\Delta F_0}{\eta T} + \eta \sigma_{alt,1}^2 + \eta^2 \sigma_{alt,2}^2. \tag{43}$$

**Corollary 3.** *An optimal learning rate is chosen as follows*

$$\eta = \left(\frac{\Delta F_0}{T \sigma_{alt,1}^2}\right)^{1/2} \bigwedge \left(\frac{\Delta F_0^2}{T^2 \sigma_{alt,2}^2}\right)^{1/3} \bigwedge \frac{1}{1 + \mu^2} \bigwedge \frac{m}{\chi^2(n-m)} \bigwedge \sqrt{\frac{m}{\chi^2 n}}. \tag{44}$$

*We have, ignoring absolute constants,*

$$\frac{1}{T} \sum_{t=0}^{T-1} \left( \frac{1}{L_{\mathbf{u}}} \mathbb{E}\left[\Delta_{\mathbf{u}}^{(t)}\right] + \frac{m}{nL_{\mathbf{v}}} \mathbb{E}\left[\Delta_{\mathbf{v}}^{(t)}\right] \right) \leq \tag{45}$$

$$\frac{(\Delta F_0 \sigma_{alt,1}^2)^{1/2}}{\sqrt{T}} + \frac{(\Delta F_0^2 \sigma_{alt,2}^2)^{1/3}}{T^{2/3}} + \frac{\Delta F_0}{T} \left(1 + \mu^2 + \chi^2(\frac{n}{m} - 1) + \sqrt{\chi^2 \frac{n}{m}}\right). \tag{46}$$

*Proof.* The proof is invoking Lemma 25 in Pillutla et al. (2022) upon establishing Thm. 1 □

We will refer the readers to Pillutla et al. (2022) for the proof of convergence of FedAlt algorithm therein. One of our novel differences to Pillutla et al. (2022) is that our black-box model is secure so its gradient is invisible to us, leading us to consider zero-order (gradient free) optimization. Thus we first establish a result analogous to lemma 22 in Pillutla et al. (2022) for zero-order optimization.

**Lemma 2.** *Consider $f : R^d \rightarrow R$ which is L-smooth, its norm of gradient is bounded by $M$ and fix a $\mathbf{w}^{(0)} \in R^d$. Define the sequence $(\mathbf{w}^{(t)})$ of iterates produced by stochastic difference descent with step $\rho$ and a fixed learning rate $\gamma$ starting from $\mathbf{w}^{(0)}$:*

$$\mathbf{w}^{(t+1)} = \mathbf{w}^{(t)} - \gamma \tilde{g}^{(t)}, \tag{47}$$

*where $\tilde{g}$ is an unbiased estimation to $\Delta_{\rho, \mathbf{w}} f(\mathbf{w}^{(t)})$ (not an unbiased estimation to $\nabla f(\mathbf{w}^{(t)})$) with bounded variance $2M^4 L^2 \rho^2 + 2\sigma^2$. Fix a number $\tau$ of steps. If $\gamma \leq (\sqrt{2}\tau L)^{-1}$, we have the bound*

$$\sum_{t=0}^{\tau-1} ||\mathbf{w}^{(t)} - \mathbf{w}^{(0)}||^2 \leq 8\gamma^2 \tau^2 (\tau - 1) ||\nabla f(\mathbf{w}^{(0)})||^2 + 4\gamma^2 \tau^2 (\tau - 1)(2M^4 L^2 \rho^2 + 2\sigma^2). \tag{48}$$

*Proof.* If $\tau = 1$, we have nothing to prove. Assume now that $\tau \geq 2$. Let $\mathcal{F}^{(t)}$ be the sigma-algebra generated by $\mathbf{w}^{(t)}$ and denote $\mathbb{E}_t[\cdot] = \mathbb{E}[\cdot | \mathcal{F}^{(t)}]$. We will use the inequality

$$\mathbb{E}_t ||\tilde{g}^{(t)}||^2 = \mathbb{E}_t ||\tilde{g}^{(t)} - \nabla f(\mathbf{w}^{(t)})||^2 + ||\nabla f(\mathbf{w}^{(t)})||^2 \leq 2M^4 L^2 \rho^2 + 2\sigma^2 + ||\nabla f(\mathbf{w}^{(t)})||^2. \tag{49}$$

We then successively deduce,

$$\mathbb{E}_t ||\mathbf{w}^{(t+1)} - \mathbf{w}^{(0)}||^2 = ||\mathbf{w}^{(t)} - \mathbf{w}^{(0)} - \gamma \tilde{g}^{(t)}||^2 \tag{50}$$

$$\leq \left(1 + \frac{1}{\tau - 1}\right) ||\mathbf{w}^{(t)} - \mathbf{w}^{(0)}||^2 + \gamma^2 \tau \mathbb{E}_t ||\tilde{g}^{(t)}||^2 \tag{51}$$

$$\leq \left(1 + \frac{1}{\tau - 1}\right) ||\mathbf{w}^{(t)} - \mathbf{w}^{(0)}||^2 + 2\gamma^2 \tau ||\nabla f(\mathbf{w}^{(t)} - \nabla f(\mathbf{w}^{(0)})||^2 \tag{52}$$

$$+ 2\gamma^2 \tau ||\nabla f(\mathbf{w}^{(0)})||^2 + \gamma^2 \tau (2M^4 L^2 \rho^2 + 2\sigma^2) \tag{53}$$

$$\leq \left(1 + \frac{1}{\tau - 1} + 2\gamma^2 \tau L^2\right) ||\mathbf{w}^{(t)} - \mathbf{w}^{(0)}||^2 + 2\gamma^2 \tau ||\nabla f(\mathbf{w}^{(0)})||^2 \tag{54}$$

$$+ \gamma^2 \tau (2M^4 L^2 \rho^2 + 2\sigma^2) \tag{55}$$

$$\leq \left(1 + \frac{2}{\tau - 1}\right) ||\mathbf{w}^{(t)} - \mathbf{w}^{(0)}||^2 + 2\gamma^2 \tau ||\nabla f(\mathbf{w}^{(0)})||^2 \tag{56}$$

$$+ \gamma^2 \tau (2M^4 L^2 \rho^2 + 2\sigma^2). \tag{57}$$

Above, we used (a) the inequality $2\alpha\beta \leq \alpha^2/\delta^2 + \delta^2 \beta^2$ for reals $\alpha, \beta, \delta$, (b) equation 49, (c) $L$-smoothness of $f$, and, (d) the condition on the learning rate.

Let $C = 2\gamma^2 \tau ||\nabla f(\mathbf{w}^{(0)})||^2 + \gamma^2 \tau (2M^4 L^2 \rho^2 + 2\sigma^2)$. Unrolling the inequality and summing up the series for all $t \leq \tau - 1$

$$||\mathbf{w}^{(t)} - \mathbf{w}^{(0)}||^2 \leq C \sum_{j=0}^{t-1} \left(1 - \frac{2}{\tau - 1}\right)^j \leq \frac{C}{2}(\tau - 1)\left(1 + \frac{2}{\tau - 1}\right)^t \tag{58}$$

$$\leq \frac{C}{2}(\tau - 1)\left(1 + \frac{2}{\tau - 1}\right)^{\tau - 1} \leq \frac{C}{2}(\tau - 1)e^2, \tag{59}$$

where we used the bound $(1 + 1/\alpha)^\alpha \leq e$ for all $\alpha > 0$. Summing over $t$ and using the numerical bound $e^2 < 8$ completes the proof.

$\square$

**Lemma 3.** *Consider the setting of Lemma. 2. If $\gamma \leq (2\tau L)^{-1}$, we have the bound*

$$||\mathbf{w}^{(\tau)} - \mathbf{w}^{(0)}||^2 \leq 16\gamma^2 \tau^2 ||\nabla f(\mathbf{w}^{(0)})||^2 + 8\gamma^2 \tau^2 (2M^4 L^2 \rho^2 + 2\sigma^2). \tag{60}$$

*Proof.* Proceeding similar to the last proof (expect using $\delta = \tau$) gives us

$$\mathbb{E}_t ||\mathbf{w}^{(t+1)} - \mathbf{w}^{(0)}||^2 \leq \left(1 + \frac{2}{\tau}\right) ||\mathbf{w}^{(t)} - \mathbf{w}^{(0)}||^2 + 4\gamma^2 \tau ||\nabla f(\mathbf{w}^{(0)})||^2 + 2\gamma^2 \tau (2M^4 L^2 \rho^2 + 2\sigma^2). \tag{61}$$

Unrolling and summing up the sequence completes the proof, similar to that of Lemma. 2 $\square$

With all the preparations, we now take the proof of Thm. 1.

*Proof.* Recall that our parameters are renamed as $\mathbf{u} := \hat{\mathbf{u}} \cup \mathbf{v}_1, \mathbf{v} := \mathbf{v}_2$.

The first step is to start with

$$F(\mathbf{u}^{(t+1)}, \mathbf{v}^{(t+1)}) - F(\mathbf{u}^{(t)}, \mathbf{v}^{(t)}) = F(\mathbf{u}^{(t)}, \mathbf{v}^{(t+1)}) - F(\mathbf{u}^{(t)}, \mathbf{v}^{(t)}) \tag{62}$$

$$+ F(\mathbf{u}^{(t+1)}, \mathbf{v}^{(t+1)}) - F(\mathbf{u}^{(t)}, \mathbf{v}^{(t+1)}). \tag{63}$$

The first line is referred to $\mathbf{v}$-step and the second $\mathbf{u}$-step. The smoothness assumption bounds the $\mathbf{u}$-step:

$$F(\mathbf{u}^{(t+1)}, \mathbf{v}^{(t+1)}) - F(\mathbf{u}^{(t)}, \mathbf{v}^{(t+1)}) = \frac{1}{n} \sum_{i=1}^{n} F_i(\mathbf{u}_i^{(t+1)}, \mathbf{v}_i^{(t+1)}) - F_i(\mathbf{u}_i^{(t)}, \mathbf{v}_i^{(t)}) \tag{64}$$

$$\leq \frac{1}{n} \sum_{i=1}^{n} \left( \left\langle \nabla_{\mathbf{u}} F_i(\mathbf{u}_i^{(t)}, \mathbf{v}_i^{(t+1)}), \mathbf{u}_i^{(t+1)} - \mathbf{u}_i^{(t)} \right\rangle + \frac{L_{\mathbf{u}}}{2} ||\mathbf{u}_i^{(t+1)} - \mathbf{u}_i^{(t)}||^2 \right). \tag{65}$$

As discussed in Pillutla et al. (2022), the most challenging thing is that the two terms in angle bracket $\langle \rangle$ are not independent random variables. Indeed, they both depend on the sampling $S^{(t)}$ of devices. The way to circumvent it is to introduce virtual full participation for $\mathbf{v}$-step update to eliminate this dependence structure and obtain a good estimate of the error it introduces. Briefly speaking, virtual full participation for parameters $\mathbf{v}$ is to assume all devices to update the $\mathbf{v}$ parameters (it is just technically assumed but not done in practice) that is independent of sampling of $S^{(t)}$ devices, breaking the dependence between $\mathbf{u}^{(t+1)}$ and $\mathbf{v}^{(t+1)}$. We ask the readers to read Pillutla et al. (2022) for full details.

We use the notation $\tilde{\mathbf{v}}^{(t+1)}$ to denote the virtual update of $\mathbf{v}$. Then the proceeding inequality goes on as:

$$\left\langle \nabla_{\mathbf{u}} F_i(\mathbf{u}_i^{(t)}, \mathbf{v}_i^{(t+1)}), \mathbf{u}_i^{(t+1)} - \mathbf{u}_i^{(t)} \right\rangle + \frac{L_{\mathbf{u}}}{2} ||\mathbf{u}_i^{(t+1)} - \mathbf{u}_i^{(t)}||^2 \tag{66}$$

$$= \left\langle \nabla_{\mathbf{u}} F_i(\mathbf{u}_i^{(t)}, \tilde{\mathbf{v}}_i^{(t+1)}), \mathbf{u}_i^{(t+1)} - \mathbf{u}_i^{(t)} \right\rangle + \frac{L_{\mathbf{u}}}{2} ||\mathbf{u}_i^{(t+1)} - \mathbf{u}_i^{(t)}||^2 \tag{67}$$

$$+ \left\langle \nabla_{\mathbf{u}} F_i(\mathbf{u}_i^{(t)}, \mathbf{v}_i^{(t+1)}) - \nabla_{\mathbf{u}} F_i(\mathbf{u}_i^{(t)}, \tilde{\mathbf{v}}_i^{(t+1)}), \mathbf{u}_i^{(t+1)} - \mathbf{u}_i^{(t)} \right\rangle \tag{68}$$

$$\leq \left\langle \nabla_{\mathbf{u}} F_i(\mathbf{u}_i^{(t)}, \tilde{\mathbf{v}}_i^{(t+1)}), \mathbf{u}_i^{(t+1)} - \mathbf{u}_i^{(t)} \right\rangle + L_{\mathbf{u}} ||\mathbf{u}_i^{(t+1)} - \mathbf{u}_i^{(t)}||^2 \tag{69}$$

$$+ \frac{1}{2L_{\mathbf{u}}} ||\nabla_{\mathbf{u}} F_i(\mathbf{u}_i^{(t)}, \mathbf{v}_i^{(t+1)}) - \nabla_{\mathbf{u}} F_i(\mathbf{u}_i^{(t)}, \tilde{\mathbf{v}}_i^{(t+1)})||^2 \tag{70}$$

$$\leq \left\langle \nabla_{\mathbf{u}} F_i(\mathbf{u}_i^{(t)}, \tilde{\mathbf{v}}_i^{(t+1)}), \mathbf{u}_i^{(t+1)} - \mathbf{u}_i^{(t)} \right\rangle + L_{\mathbf{u}} ||\mathbf{u}_i^{(t+1)} - \mathbf{u}_i^{(t)}||^2 \tag{71}$$

$$+ \frac{\chi^2 L_{\mathbf{v}}}{2} ||\tilde{\mathbf{v}}^{(t+1)} - \mathbf{v}_i^{(t+1)}||^2. \tag{72}$$

The last two inequalities follow from Young's inequality and Lipschitzness of $\mathbf{v}_i \hookrightarrow \nabla_{\mathbf{u}} F_i(\mathbf{u}_i, \mathbf{v}_i)$ respectively.

The usage of virtual update is to ensure that $\tilde{\mathbf{v}}^{(t+1)}$ is independent of $S^{(t)}$. This allows us to take an expectation w.r.t the sampling $S^{(t)}$ of the devices.

Recall that under the new parameters $\mathbf{u}$ and $\mathbf{v}$, the only difference to the setting of FedAlt in Pillutla et al. (2022) is that we need zero-order optimization to update $\mathbf{u}$ parameter instead of first-order gradient method. Thus, we can use the calculations in Pillutla et al. (2022) with $F$ replaced by $F_i$

and then add them together from 1 to $n$ to similarly arrive at the following expression:

$$\mathbb{E}_t\left[F(\mathbf{u}^{(t+1)}, \mathbf{v}^{(t+1)}) - F(\mathbf{u}^{(t)}, \mathbf{v}^{(t+1)})\right] \tag{73}$$

$$\le -\frac{\gamma_\mathbf{u}\tau_\mathbf{u}}{4}\mathbb{E}_t[\tilde{\Delta}_\mathbf{u}^{(t)}] + \frac{2\gamma_\mathbf{u}L_\mathbf{u}^2}{n}\sum_{i=1}^{n}\sum_{k=0}^{\tau_\mathbf{u}-1}\mathbb{E}_t||\tilde{\mathbf{u}}_{i,k}^{(t)} - \mathbf{u}^{(t)}||^2 \tag{74}$$

$$+ 4\gamma_\mathbf{v}^2\tau_\mathbf{v}^2 L_\mathbf{v}(2M_\mathbf{v}^4 L_\mathbf{v}^2\rho^2 + 2\sigma_\mathbf{v}^2)\chi^2(1 - m/n) \tag{75}$$

$$+ \frac{L_\mathbf{u}\gamma_\mathbf{u}^2\tau_\mathbf{u}^2}{m}(2M_\mathbf{u}^4 L_\mathbf{u}^2\rho^2 + 2\sigma_\mathbf{u}^2 + 3\delta^2(1 - \frac{m}{n})) + 8\gamma_\mathbf{v}^2\tau_\mathbf{v}^2 L_\mathbf{v}\chi^2(1 - m/n)\Delta_\mathbf{v}^{(t)}, \tag{76}$$

note here $\tilde{\mathbf{u}}_{i,k}^{(t)}$ is $\mathbf{u}$-parameter updates via stochastic difference descent method rather than stochastic gradient descent method. We bound this term with Lemma. 2, invoking the assumption on gradient diversity. And then plugging the resulting estimate back in, we get

$$\mathbb{E}_t\left[F(\mathbf{u}^{(t+1)}, \mathbf{v}^{(t+1)}) - F(\mathbf{u}^{(t)}, \mathbf{v}^{(t+1)})\right] \tag{77}$$

$$\le -\frac{\gamma_\mathbf{u}\tau_\mathbf{u}}{8}\mathbb{E}_t[\tilde{\Delta}_\mathbf{u}^{(t)}] + \frac{L_\mathbf{u}\gamma_\mathbf{u}^2\tau_\mathbf{u}^2}{m}(2M_\mathbf{u}^4 L_\mathbf{u}^2\rho^2 + 2\sigma_\mathbf{u}^2 + 2\delta^2(1 - m/n)) \tag{78}$$

$$+ 4\gamma_\mathbf{v}^2\tau_\mathbf{v}^2 L_\mathbf{v}(2M_\mathbf{v}^4 L_\mathbf{v}^2\rho^2 + 2\sigma_\mathbf{v}^2)\chi^2(1 - m/n) \tag{79}$$

$$+ 8\gamma_\mathbf{v}^2\tau_\mathbf{v}^2 L_\mathbf{v}\chi^2(1 - m/n)\Delta_\mathbf{v}^{(t)} + 8\gamma_\mathbf{v}^2 L_\mathbf{u}^3\tau_\mathbf{u}^2(\tau_\mathbf{u} - 1)(2M_\mathbf{u}^4 L_\mathbf{u}^2\rho^2 + 2\sigma_\mathbf{u}^2 + 2\delta_\mathbf{u}^2). \tag{80}$$

The bound on $\mathbf{v}$-step has exactly the same form as presented in Pillutla et al. (2022) since when conditioned on $\mathbf{u}^{(t)}, \mathbf{v}^{(t)}$ all functions used in updating $\mathbf{v}$ is continuously differentiable. Plugging this bound on $\mathbf{v}$-step into equation 62, we get

$$\mathbb{E}_t\left[F(\mathbf{u}^{(t+1)}, \mathbf{v}^{(t+1)} - F(\mathbf{u}^{(t)}, \mathbf{v}^{(t)})\right] \tag{81}$$

$$\le -\frac{\gamma_\mathbf{u}\tau_\mathbf{u}}{8}\mathbb{E}_t[\tilde{\Delta}_\mathbf{u}^{(t)}] - \frac{\gamma_\mathbf{v}\tau_\mathbf{v}m}{16n}\mathbb{E}_t[\Delta_\mathbf{v}^{(t)}] + 4\gamma_\mathbf{v}^2 L_\mathbf{v}\tau_\mathbf{v}^2(2M_\mathbf{v}^4 L_\mathbf{v}^2\rho^2 + 2\sigma_\mathbf{v}^2)\left(\frac{m}{n} + \chi^2(1 - m/n)\right) \tag{82}$$

$$+ \frac{\gamma_\mathbf{u}^2 L_\mathbf{u}\tau_\mathbf{u}^2}{m}(2M_\mathbf{u}^4 L_\mathbf{u}^2\rho^2 + 2\sigma_\mathbf{u}^2 + 2\delta^2(1 - m/n)) + 8\gamma_\mathbf{u}^3 L_\mathbf{u}^2\tau_\mathbf{u}^2(\tau_\mathbf{u} - 1)(2M_\mathbf{u}^4 L_\mathbf{u}^2\rho^2 + 2\sigma_\mathbf{u}^2 + 2\delta^2) \tag{83}$$

$$+ \frac{4\gamma_\mathbf{v}^3 L_\mathbf{v}^2\tau_\mathbf{v}^2(\tau_\mathbf{v} - 1)(2M_\mathbf{v}^4 L_\mathbf{v}^2\rho^2 + 2\sigma_\mathbf{v}^2)m}{n}. \tag{84}$$

Taking an unconditional expectation, summing it over $t = 0$ to $T - 1$ and rearranging this gives

$$\frac{1}{T} = \sum_{t=0}^{T-1}\left(\frac{\gamma_\mathbf{u}\tau_\mathbf{u}}{8}\mathbb{E}[\tilde{\Delta}_\mathbf{u}^{(t)}] + \frac{\gamma_\mathbf{v}\tau_\mathbf{v}m}{16}\mathbb{E}[\Delta_\mathbf{v}^{(t)}]\right) \tag{85}$$

$$\le \frac{\Delta F_0}{T} + 4\gamma_\mathbf{v}^2 L_\mathbf{v}\tau_\mathbf{v}^2(2M_\mathbf{v}^4 L_\mathbf{v}^2\rho^2 + 2\sigma_\mathbf{v}^2)\left(\frac{m}{n} + \chi^2(1 - m/n)\right) \tag{86}$$

$$+ \frac{\gamma_\mathbf{u}^2 L_\mathbf{u}\tau_\mathbf{u}^2}{m}(2M_\mathbf{u}^4 L_\mathbf{u}^2\rho^2 + 2\sigma_\mathbf{u}^2 + 2\delta^2(1 - m/n)) \tag{87}$$

$$+ 8\gamma_\mathbf{u}^3 L_\mathbf{u}^2\tau_\mathbf{u}^2(\tau_\mathbf{u} - 1)(2M_\mathbf{u}^4 L_\mathbf{u}^2\rho^2 + 2\sigma_\mathbf{u}^2 + 2\delta^2) + \frac{4\gamma_\mathbf{v}^3 L_\mathbf{v}^2\tau_\mathbf{v}^2(\tau_\mathbf{v} - 1)(2M_\mathbf{v}^4 L_\mathbf{v}^2\rho^2 + 2\sigma_\mathbf{v}^2)m}{n}. \tag{88}$$

This is a bound in terms of the virtual updates $\tilde{\mathbf{v}}^{(t+1)}$. Similarly to the manipulations in Pillutla et al. (2022) we can relate $\tilde{\Delta}_{\mathbf{u}}^{(t)}$ with $\Delta_{\mathbf{u}}^{(t)}$. [9], and finally we get:

$$\frac{1}{T}\sum_{t=0}^{T-1}\left(\frac{\gamma_{\mathbf{u}}\tau_{\mathbf{u}}}{16}\mathbb{E}[\Delta_{\mathbf{u}}^{(t)}] + \frac{\gamma_{\mathbf{v}}\tau_{\mathbf{v}}m}{32n}\mathbb{E}[\Delta_{\mathbf{v}}^{(t)}]\right) \tag{89}$$

$$\leq \frac{1}{T}\sum_{t=0}^{T-1}\left(\frac{\gamma_{\mathbf{u}}\tau_{\mathbf{u}}}{9}\mathbb{E}[\tilde{\Delta}_{\mathbf{u}}^{(t)}] + \frac{\gamma_{\mathbf{v}}\tau_{\mathbf{v}}m}{16n}\mathbb{E}[\Delta_{\mathbf{v}}^{(t)}]\right) + \gamma_{\mathbf{u}}\tau_{\mathbf{u}}\gamma_{\mathbf{v}}^2\tau_{\mathbf{v}}^2(2M_{\mathbf{u}}^4 L_{\mathbf{u}}^2\rho^2 + 2\sigma_{\mathbf{u}}^2)\chi^2 L_{\mathbf{u}}L_{\mathbf{v}} \tag{90}$$

$$\leq \frac{\Delta F_0}{T} + 4\gamma_{\mathbf{v}}^2 L_{\mathbf{v}}\tau_{\mathbf{v}}^2(2M_{\mathbf{v}}^4 L_{\mathbf{v}}^2\rho^2 + 2\sigma_{\mathbf{v}}^2)\left(\frac{m}{n} + \chi^2(1 - m/n)\right) \tag{91}$$

$$+ \frac{\gamma_{\mathbf{u}}^2 L_{\mathbf{u}}\tau_{\mathbf{u}}^2}{m}(2M_{\mathbf{u}}^4 L_{\mathbf{u}}^2\rho^2 + 2\sigma_{\mathbf{u}}^2 + 2\delta^2(1 - m/n)) \tag{92}$$

$$+ 8\gamma_{\mathbf{u}}^3 L_{\mathbf{u}}^2\tau_{\mathbf{u}}^2(\tau_{\mathbf{u}} - 1)(2M_{\mathbf{u}}^4 L_{\mathbf{u}}^2\rho^2 + 2\sigma_{\mathbf{u}}^2 + 2\delta^2) + \frac{4\gamma_{\mathbf{v}}^3 L_{\mathbf{v}}^2\tau_{\mathbf{v}}^2(\tau_{\mathbf{v}} - 1)(2M_{\mathbf{v}}^4 L_{\mathbf{v}}^2\rho^2 + 2\sigma_{\mathbf{v}}^2)m}{n} \tag{93}$$

$$+ \gamma_{\mathbf{v}}\tau_{\mathbf{u}}\gamma_{\mathbf{v}}^2\tau_{\mathbf{v}}^2(2M_{\mathbf{v}}^4 L_{\mathbf{v}}^2\rho^2 + 2\sigma_{\mathbf{v}}^2)\chi^2 L_{\mathbf{u}}L_{\mathbf{v}}. \tag{94}$$

Plugging in $\gamma_{\mathbf{u}} = \eta/(L_{\mathbf{u}}\tau_{\mathbf{u}})$ and $\gamma_{\mathbf{v}} = \eta/(L_{\mathbf{v}}\tau_{\mathbf{v}})$ completes the proof.

$\square$

## D  MORE DISCUSSION ON RELATED WORK

**Federated learning** makes it possible to perform distributed multi-party computing without comprising privacy (Zhang et al., 2021; Voigt & Von dem Bussche, 2017; McMahan et al., 2017; Yang et al., 2019; Wan et al., 2023; Qi et al., 2023). FedAVG is the baseline algorithm for FL by exchanging parameters instead of raw data, which has been used in many applications (Li et al., 2020a; Banabilah et al., 2022; Rodríguez-Barroso et al., 2023). When FedAVG meets non-iid data, it can suffer from low convergence speed and terrible **personalization** performance (Sattler et al., 2019). FedProx (Li et al., 2020b) allowed differences among clients and the server while FedBN (Li et al., 2021) preserved local batch normalization layers in each client. Setayesh et al. (2023) proposed PerFedMask, a generalization of FedBABU (Oh et al., 2022), that considers the computational capability of different devices. Xu et al. (2023b) conducted explicit local-global feature alignment by leveraging global semantic knowledge, quantified the benefit of classifier combination for each client, and derived an optimization problem for estimating optimal weights. Some other work considered utilizing knowledge distillation for personalization (Chen et al., 2023c) while some work attempted to achieve robust personalization during testing (Jiang & Lin, 2023). Besides personalization, there also exists research focusing on generalization (Chen & Chao, 2022; Gupta et al., 2022; Qu et al., 2022). Our method can deal with situations where distribution shifts exist.

Since deep learning has entered the era of **large foundation models** (Bommasani et al., 2021; Xing et al., 2023; Zhuang et al., 2023), some novel issues, e.g. computation costs and communication costs, are coming into being, leading operations on the whole network impossible (Wang et al., 2022; Chen et al., 2022; 2023a; Ding et al., 2023; Xiao et al., 2023). FedPrompt (Zhao et al., 2023) studied prompt tuning in a model split aggregation way using FL while FedCLIP (Lu et al., 2023) designed an attention-based adapter for CLIP (Radford et al., 2021). PromptFL (Guo et al., 2023) utilized the federated prompt training instead of the whole model training in a distributed way while pFedPG (Yang et al., 2023a) deployed a personalized prompt generator at the server to produce client-specific visual prompts. FwdLLM (Xu et al., 2023b) combined BP-free training with parameter-efficient training methods and systematically and adaptively allocated computational loads across devices. These methods all require access to the internals of large models but we view foundation models as black-box in this paper.

Besides data privacy, **model privacy** also raised attention recently (Mo et al., 2020). Model suppliers are usually more willing to only provide predictions for given inputs or just provide a product that can only generate predictions (Van Dis et al., 2023). In this paper, we view these protected foundation models as black-box models (Guidotti et al., 2018; Ljung, 2001). Little work paid attention to

---

[9]More precisely, we simply take the same steps for $F_i$ and then add all of them together.

finetuning or optimizing in this field, but most related work focused on attacks (Yang et al., 2023b;c). One related work is FedZO (Fang et al., 2022) which utilized **zero-order optimization** (Ghadimi & Lan, 2013), but it did not consider utilizing large foundation models. Some other work also made use of zero-order optimization for federated learning (Li & Chen, 2021; Zelikman et al., 2023; Feng et al., 2023), but none of them utilized large black-box foundation models.

**Model reprogramming** (MR) (Tsai et al., 2020; Xu et al., 2023c; Chen, 2022) provides a similar solution to ZOOPFL. It trains the inserted input transformation and output mapping layers while keeping the source pretrained model inact to enable resource-efficient cross-domain machine learning. The main purpose of model reprogramming is to transfer knowledge to targets and it can be viewed as a sub-field of transfer learning. Recently, Arif et al. (2023) proposed the first framework, Reprogrammable-FL, adapting MR to the setting of differentially private federated learning. Reprogrammable-FL learned an input transformation for samples and added learned perturbations to the original samples. It preserved local input transformations and shared output transformation layers, which are totally in contrast to ours. Moreover, ZOOPFL is proposed for black-box foundation models and can provide an ideal personalization capability.

# E   METHODOLOGY

## E.1   ALGORITHM FLOW

Algorithm 1 describes the concrete process of our proposed ZOOPFL. $s_i$ is consist of two components, $q_i$ (i.e. the encoder) and $o_i$ (i.e. the decoder), which are optimized via Adam in pretraining. Then, $q_i$ is optimized via CGE during Input Surgery. Finally, $r_i$ is optimized via Adam in Semantic-remapping.

---

**Algorithm 1** ZOOPFL

---

**Input**: $n$ clients' datasets $\{\mathcal{D}_i\}_{i=1}^n$
**Output**: Input surgery, $\{s_i\}_{i=1}^n$, semantic re-mapping, $\{r_i\}_{i=1}^n$, client-specific embeddings, $\{\hat{\mathbf{z}}_i\}_{i=1}^n$

1: **for** $t = 1$ to $T$ **do**                                                                                   ▷ Step 1
2:       **for** $i = 1$ to $n$ **do**
3:             Pretrain Client i according to $\ell_{MSE} = \mathbb{E}_{(\mathbf{x},y)\sim\mathbb{P}(\mathcal{D}_i)} \|o_i([q_i(\mathbf{x}), \hat{\mathbf{z}}_i]) - \mathbf{x}\|_2^2$
4:       Upload $\{w(s_i)\}_{i=1}^n$ to the server
5:       Aggregate weights, $w(s) = \frac{1}{n}\sum_{i=1}^n w(s_i)$
6:       Distribute client weight $w(s)$ to each client

7: **for** $t = 1$ to $T$ **do**
8:       **for** $i = 1$ to $n$ **do**                                                                            ▷ Step 2
9:             **for** $k = 1$ to $d_1 + d_2$ **do**                                     ▷ Compute Gradients via CGE
10:                   $\tilde{\mathbf{z}}_1 = \tilde{\mathbf{z}} + \rho\mathbf{e}_j, \ell_{\mathbf{x},1} = \ell_{cls}(r_i(g(o_i(\tilde{\mathbf{z}}_1, y))))$
11:                   $\tilde{\mathbf{z}}_2 = \tilde{\mathbf{z}} - \rho\mathbf{e}_j, \ell_{\mathbf{x},2} = \ell_{cls}(r_i(g(o_i(\tilde{\mathbf{z}}_2, y))))$
12:                   Compute differences on embeddings, $\nabla_{\tilde{\mathbf{z}}}\mathcal{G}(\tilde{\mathbf{z}}) \approx \sum_{i=1}^{d_1+d_2} \frac{\ell_{\mathbf{x},2} - \ell_{\mathbf{x},1}}{2\times\rho}\mathbf{e}_j$
13:                   Compute gradients, $\nabla_{q_i}\ell_1 = \frac{d\tilde{\mathbf{z}}}{dq_i}\frac{d\mathcal{G}}{d\tilde{\mathbf{z}}} \approx \frac{d\mathbf{z}}{dq_i}\nabla_{\tilde{\mathbf{z}}}\mathcal{G}(\tilde{\mathbf{z}})_1 \approx \frac{d\nabla_{\tilde{\mathbf{z}}}\mathcal{G}(\tilde{\mathbf{z}})_1\mathbf{z}}{dq_i}$
14:                   Update parameters, $w(q_i^{new}) = w(q_i) - \gamma_1 \times \nabla_{q_i}\ell_1$
15:       Upload $\{w(q_i^{new})\}_{i=1}^n$ to the server
16:       Aggregate weights, $w(q) = \frac{1}{n}\sum_{i=1}^n w(q_i^{new})$
17:       Distribute $q$ to clients
18:       **for** $i = 1$ to $n$ **do**                                                                          ▷ Step 3
19:             Train semantic re-mapping according to $\ell_2 = \mathbb{E}_{(\mathbf{x},y)\sim\mathbb{P}(\mathcal{D}_i)}\ell_{cls}(r_i(\mathcal{F}(\mathbf{x})), y)$

---

# F   EXPERIMENTS

## F.1   DESCRIPTION OF THE DATASETS

**Computer vision datasets.**

**COVID-19 (Sait et al., 2020).** It is a public posterior-anterior chest radiography images dataset with four classes, including 1,281 COVID-19 X-Rays, 3,270 Normal X-Rays, 1,656 viral-pneumonia X-Rays, and 3,001 bacterial-pneumonia X-Rays. We split data into 20 clients via the Dirichlet distribution following (Yurochkin et al., 2019) and each client has different distributions on the label space. In each client, only $10\%$ of data are utilized for training and the rest data are split evenly into two parts for validation and testing respectively.

**APTOS (Karthik, 2019).** It is an image dataset that judges the severity of diabetic retinopathy on a scale of 0 to 4. The original dataset contains 3,662 training images and 1,928 testing images but it suffers from heavily unbalanced. We only utilize the training part in our setting and we randomly choose 400 samples for classes 0 and 2. Our processed dataset contains 1658 samples. We split data into 20 clients and $20\%$ of data serve for training similar to COVID-19.

**Terra Incognita (Beery et al., 2018).** It is a common dataset that contains photographs of wild animals taken by camera traps at locations L100, L38, L43, and L46. It totally contains 24,788 samples with 10 classes. We randomly choose two locations, i.e. L46 and L100, to construct two benchmarks, i.e. Terra46 and Terra100. For these two benchmarks, we split data into 20 clients and $20\%$ of data serve for training similar to COVID-19.

**Natural language processing datasets.**

**SST-2 (Wang et al., 2019; Socher et al., 2013).** The Stanford Sentiment Treebank contains sentences from movie reviews and the labels come from human annotations of their sentiment. It contains about 67k training samples with two classes. We choose sentences with the number of words ranging from 20 to 50 [10] and obtain 9763 samples. We split data into 20 clients and $20\%$ of data serve for training similar to COVID-19.

**COLA (Wang et al., 2019; Warstadt et al., 2019).** The Corpus of Linguistic Acceptability contains English acceptability judgments drawn from books and journal articles on linguistic theory and it judges a sequence of words whether it is a grammatical English sentence. It contains 8.5k training samples with two classes. We choose sentences with the number of words less than 30 and obtain 5700 samples in total. For the class with more samples, we randomly choose parts to ensure balance. We split data into 20 clients and $20\%$ of data serve for training similar to COVID-19.

**Financial-phrasebank (Malo et al., 2014).** It consists of sentences from financial news categorized by sentiment. It contains 4840 samples with three classes, including positive, neutral, and negative. We choose sentences with the number of words less than 60 and obtain 3379 samples in total. We split data into 10 clients and $20\%$ of data serve for training similar to COVID-19.

**Flipkart (Vaghani & Thummar, 2023).** This dataset contains information on product name, product price, rate, reviews, summary and sentiment. It has 205,053 samples with multiple labels. We choose sentiment analysis as our task and there can be three classes, including positive, neutral, and negative. We choose reviews with lengths more than 30 and we randomly choose parts of classes with more samples for balance. The processed dataset contains 3048 samples for each class. We split data into 20 clients and $20\%$ of data serve for training similar to COVID-19.

### F.2  Data Distributions

Figure 7 shows the complete data distributions on the rest benchmarks.

### F.3  Model Structures

Table 4 shows details on auto-encoders. Please note that the encoder accounts for approximately half of the parameter count.

### F.4  Additional Results

Table 5, Table 6, and Table 7 show more detailed results on computer vision and natural language processing.

---

[10]We split sentences into words via spaces.

Table 4: The structures of auto-encoders.

| CV | | | NLP(Linear) | | |
|---|---|---|---|---|---|
| Layer(type) | Output Shape | #—Param— | Layer(type) | Output Shape | #—Param— |
| Conv2d-1 | [-1,32,224,224] | 896 | Linear-1 | [-1,128] | 6,291,584 |
| ReLU-2 | [-1,32,224,224] | 0 | BatchNorm1d-2 | [-1,128] | 256 |
| BatchNorm2d-3 | [-1,32,224,224] | 64 | ReLU-3 | [-1,128] | 0 |
| MaxPool2d-4 | [-1,32,112,112] | 0 | Linear-4 | [-1,256] | 33,024 |
| Conv2d-5 | [-1,64,112,112] | 18,496 | BatchNorm1d-5 | [-1,256] | 512 |
| ReLU-6 | [-1,64,112,112] | 0 | ReLU-6 | [-1,256] | 0 |
| BatchNorm2d-7 | [-1,64,112,112] | 128 | Linear-7 | [-1,96] | 24,672 |
| MaxPool2d-8 | [-1,64,56,56] | 0 | BatchNorm1d-8 | [-1,96] | 192 |
| Conv2d-9 | [-1,128,56,56] | 73,856 | ReLU-9 | [-1,96] | 0 |
| ReLU-10 | [-1,128,56,56] | 0 | Linear-10 | [-1,256] | 33,024 |
| BatchNorm2d-11 | [-1,128,56,56] | 256 | BatchNorm1d-11 | [-1,256] | 512 |
| MaxPool2d-12 | [-1,128,28,28] | 0 | ReLU-12 | [-1,256] | 0 |
| Conv2d-13 | [-1,32,28,28] | 36,896 | Linear-13 | [-1,128] | 32,896 |
| ReLU-14 | [-1,32,28,28] | 0 | BatchNorm1d-14 | [-1,128] | 256 |
| BatchNorm2d-15 | [-1,32,28,28] | 64 | ReLU-15 | [-1,128] | 0 |
| MaxPool2d-16 | [-1,32,14,14] | 0 | Linear-16 | [-1,49152] | 6,340,608 |
| Conv2d-17 | [-1,8,14,14] | 2,312 | Tanh-17 | [-1,49152] | 0 |
| ReLU-18 | [-1,8,14,14] | 0 | NLP(LSTM) | | |
| BatchNorm2d-19 | [-1,8,14,14] | 16 | LSTM-1 | [-1,64,128] | 459,776 |
| MaxPool2d-20 | [-1,8,7,7] | 0 | Linear-2 | [-1,96] | 12,384 |
| Linear-21 | [-1,294] | 115,542 | LSTM-3 | [-1,64,128] | 132,096 |
| ReLU-22 | [-1,294] | 0 | Linear-4 | [-1,64,768] | 99,072 |
| BatchNorm1d-23 | [-1,294] | 588 | Tanh-5 | [-1,64,768] | 0 |
| ConvTranspose2d-24 | [-1,32,14,14] | 2,336 | | | |
| ReLU-25 | [-1,32,14,14] | 0 | | | |
| BatchNorm2d-26 | [-1,32,14,14] | 64 | | | |
| ConvTranspose2d-27 | [-1,128,28,28] | 36,992 | | | |
| ReLU-28 | [-1,128,28,28] | 0 | | | |
| BatchNorm2d-29 | [-1,128,28,28] | 256 | | | |
| ConvTranspose2d-30 | [-1,64,56,56] | 73,792 | | | |
| ReLU-31 | [-1,64,56,56] | 0 | | | |
| BatchNorm2d-32 | [-1,64,56,56] | 128 | | | |
| ConvTranspose2d-33 | [-1,32,112,112] | 18,464 | | | |
| ReLU-34 | [-1,32,112,112] | 0 | | | |
| BatchNorm2d-35 | [-1,32,112,112] | 64 | | | |
| ConvTranspose2d-36 | [-1,16,224,224] | 4,624 | | | |
| ReLU-37 | [-1,16,224,224] | 0 | | | |
| BatchNorm2d-38 | [-1,16,224,224] | 32 | | | |
| Conv2d-39 | [-1,3,224,224] | 51 | | | |

Table 5: Results on four computer vision benchmarks. **Bold** means the best.

| DataSet | Method | 1 | 2 | 3 | 4 | 5 | 6 | 7 | 8 | 9 | 10 | 11 | 12 | 13 | 14 | 15 | 16 | 17 | 18 | 19 | 20 | AVG |
|---|---|---|---|---|---|---|---|---|---|---|---|---|---|---|---|---|---|---|---|---|---|---|
| COVID-19 | Zs | 18.36 | 33.66 | 1.94 | 48.05 | 3.38 | 47.32 | 53.17 | 0.00 | 3.40 | 51.71 | 3.90 | 43.69 | 40.78 | 13.17 | 54.37 | 0.00 | 11.65 | 36.59 | 0.97 | 58.05 | 26.21 |
| | Ours | 76.81 | 42.44 | 82.04 | 59.22 | 82.61 | 70.24 | 62.44 | 79.90 | 82.04 | 55.12 | 58.54 | 80.58 | 37.86 | 71.22 | 64.08 | 82.52 | 72.33 | 49.27 | 87.38 | 66.83 | **68.17** |
| APTOS | Zs | 0.00 | 53.12 | 0.00 | 41.94 | 0.00 | 45.16 | 25.00 | 32.26 | 48.39 | 0.00 | 29.03 | 12.50 | 65.62 | 0.00 | 51.61 | 0.00 | 40.62 | 0.00 | 0.00 | 38.71 | 24.20 |
| | Ours | 43.33 | 59.38 | 54.84 | 54.84 | 50.00 | 41.94 | 56.25 | 38.71 | 51.61 | 48.39 | 41.94 | 53.12 | 65.62 | 32.26 | 48.39 | 53.12 | 50.00 | 36.67 | 48.39 | 51.61 | **49.02** |
| Terra46 | Zs | 21.93 | 21.55 | 25.22 | 12.39 | 34.48 | 6.09 | 13.04 | 25.44 | 33.33 | 18.42 | 46.09 | 6.96 | 35.96 | 16.38 | 41.74 | 5.31 | 25.00 | 7.96 | 21.74 | 21.74 | 22.04 |
| | Ours | 48.25 | 62.07 | 56.52 | 70.80 | 43.97 | 51.30 | 45.22 | 28.95 | 47.37 | 62.28 | 41.74 | 59.13 | 34.21 | 60.34 | 60.00 | 59.29 | 35.34 | 67.26 | 50.43 | 76.52 | **53.05** |
| Terra100 | Zs | 14.13 | 9.89 | 7.69 | 9.57 | 5.56 | 6.38 | 9.78 | 6.45 | 6.52 | 16.48 | 4.40 | 11.70 | 11.83 | 7.78 | 12.90 | 7.69 | 11.96 | 7.61 | 9.78 | 12.22 | 9.52 |
| | Ours | 63.04 | 73.63 | 56.04 | 75.53 | 65.56 | 100.00 | 59.78 | 96.77 | 90.22 | 57.14 | 51.65 | 64.89 | 48.39 | 80.00 | 86.02 | 49.45 | 52.17 | 68.48 | 78.26 | 60.00 | **68.85** |

Table 6: Average results on NLP. **Bold** means the best.

| Benchmark | Method | ALBERT | BERT | DeBERTa | GPT2 | AVG |
|---|---|---|---|---|---|---|
| SST-2 | ZS | 47.84 | 52.16 | 52.14 | 52.14 | 51.07 |
| | Ours | 94.70 | 94.72 | 94.70 | 94.70 | **94.71** |
| COLA | ZS | 50.22 | 50.22 | 49.87 | 48.82 | 49.78 |
| | Ours | 89.34 | 88.73 | 88.16 | 88.46 | **88.67** |
| Financial | ZS | 60.76 | 54.82 | 62.11 | 41.26 | 54.74 |
| | Ours | 68.91 | 68.54 | 68.69 | 71.79 | **69.48** |
| Flipkart | ZS | 32.26 | 29.43 | 33.27 | 34.16 | 32.28 |
| | Ours | 64.85 | 64.50 | 66.32 | 65.50 | **65.29** |

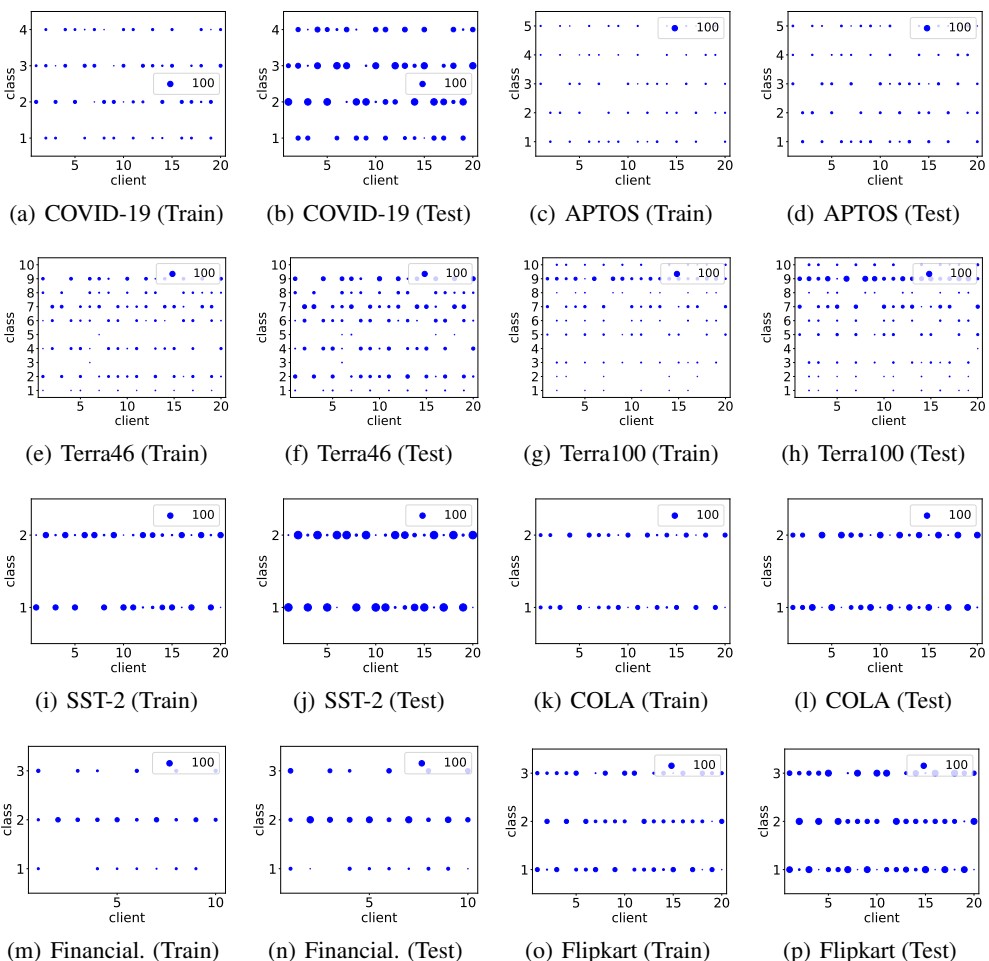

Figure 7: Data distributions. The size of dots represents the number of samples.

Table 7: Results on Financial-phrasebank. **Bold** means the best.

| Backbones | Client | 1 | 2 | 3 | 4 | 5 | 6 | 7 | 8 | 9 | 10 | AVG |
|---|---|---|---|---|---|---|---|---|---|---|---|---|
| ALBERT | ZS | 31.85 | 91.18 | 64.71 | 58.21 | 78.36 | 42.96 | 85.82 | 38.24 | 67.41 | 48.89 | 60.76 |
| | Ours | 52.59 | 99.26 | 62.50 | 58.96 | 87.31 | 52.59 | 94.78 | 50.00 | 77.78 | 53.33 | **68.91** |
| BERT | ZS | 42.22 | 70.59 | 58.82 | 49.25 | 57.46 | 47.41 | 67.16 | 43.38 | 58.52 | 53.33 | 54.82 |
| | Ours | 52.59 | 99.26 | 62.50 | 58.96 | 87.31 | 50.37 | 94.78 | 44.12 | 77.78 | 57.78 | **68.54** |
| DeBERTa | ZS | 25.93 | 99.26 | 62.50 | 58.96 | 87.31 | 36.30 | 94.78 | 31.62 | 77.78 | 46.67 | 62.11 |
| | Ours | 52.59 | 99.26 | 66.18 | 58.96 | 87.31 | 42.96 | 94.78 | 46.32 | 77.78 | 60.74 | **68.69** |
| GPT2 | ZS | 41.48 | 41.18 | 49.26 | 37.31 | 43.28 | 37.04 | 48.51 | 33.82 | 42.96 | 37.78 | 41.26 |
| | Ours | 52.59 | 99.26 | 75.00 | 61.94 | 87.31 | 52.59 | 91.04 | 50.74 | 81.48 | 65.93 | **71.79** |

## F.5    VISUALIZATION STUDY

Figure F.5 describes the visualization of both the original images and the corresponding restored images via the Autoencoder on Terra100. We can observe that the recovered images almost eliminate the corresponding category information. However, they generate distinct patterns for different categories.

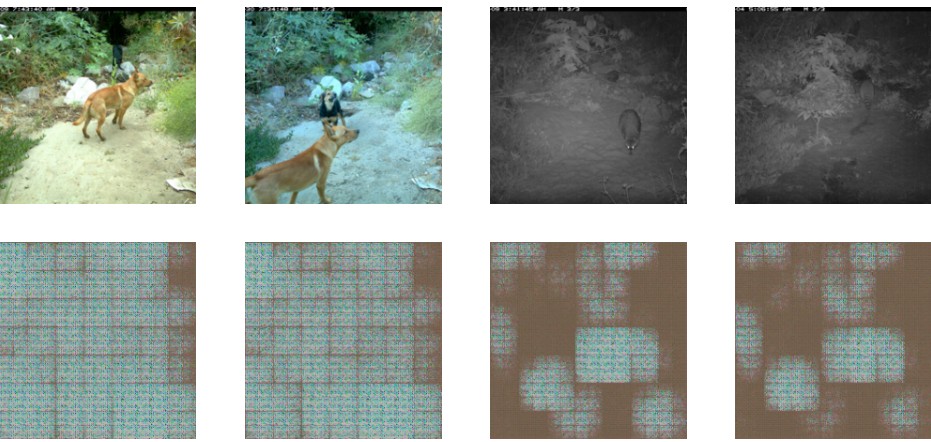

Figure 8: Visualizations of both the original images and the corresponding restored images via the Autoencoder on Terra100. The upper row is the original figures while the bottom row is the corresponding restored images via the Autoencoder. The first two columns are of the fourth class while the last two columns are of the eighth class.

