# OpenReview forum: "ZOOPFL: EXPLORING BLACK-BOX FOUNDATION MODELS FOR PERSONALIZED FEDERATED LEARNING"
_ICLR.cc/2024/Conference — Submitted to ICLR 2024_

### Official Review · Reviewer_RdGi · 2023-10-30

**Soundness:** 1 poor
**Presentation:** 3 good
**Contribution:** 1 poor
**Rating:** 3
**Confidence:** 5

**Summary:**

This paper addresses challenges in personalized federated learning with large foundation models and limited resources, including data, computation, and model access. The proposed method, ZOOPFL (Zeroth-Order Optimization for Personalized Federated Learning), adapts inputs through zeroth-order optimization and uses linear projections for personalization. Input surgery is introduced to reduce computation costs and enhance personalization.

**Strengths:**

- The experiments are comprehensive, covering multiple datasets in both computer vision (CV) and natural language processing (NLP) applications.
- The paper's focus on federated learning settings that address both data privacy and model privacy is intriguing.

**Weaknesses:**

My main concerns are the validity and privacy risks of this FL setting.
- First, the black-box FL setting lacks practicality. The paper assumes the existence of large foundation models on clients in the form of encrypted assets, and it does not require the uploading of transformed inputs. However, this does not align with the most common scenarios in machine learning model services, such as the access of various black-box large language models like ChatGPT. In practical scenarios, local data needs to be uploaded to the model service provider.
- Second, the motivation for deploying zeroth-order optimization methods based on the local encrypted black-box model setup is not well-motivated. This setting implies that it is entirely possible to train an white-box emulator [1] as a proxy for the black-box model and directly perform first-order optimization based on the white-box emulator. However, the authors do not provide relevant discussions and experimental comparisons.
- In terms of model privacy, the privacy leakage of a black-box model is closely related to the number of queries [2], but the authors do not provide theoretical or empirical studies on this.
- The experimental section lacks ablation experiments with varying levels of noise added on the transformed data and visualizations of transformed data.

[1] Xiao, Guangxuan, Ji Lin, and Song Han. "Offsite-tuning: Transfer learning without full model." arXiv preprint arXiv:2302.04870 (2023).
[2] Tsai, Yun-Yun, Pin-Yu Chen, and Tsung-Yi Ho. "Transfer learning without knowing: Reprogramming black-box machine learning models with scarce data and limited resources." International Conference on Machine Learning. PMLR, 2020.

**Questions:**

see weakness

---

> ### Author Response · Authors · 2023-11-19
> **Response to Reviewer RdGi**
>
> Thanks for your acknowledgment in our *comprehensive experiments* and *intriguing*! We see that your main concerns are on details of *practicality*, *motivation*, *model privacy*, and *more experimental analysis*. Now we answer them here.
>
> 1. The black-box FL setting lacks practicality.
>
> Please refer to the general response for the problem setting.
>
> 2.  The motivation for deploying zeroth-order optimization methods based on the local encrypted black-box model setup is not well-motivated. This setting implies that it is entirely possible to train an white-box emulator [1] as a proxy for the black-box model and directly perform first-order optimization based on the white-box emulator. However, the authors do not provide relevant discussions and experimental comparisons.
>
> It is important to note that we are in a black-box setting while [1] uses a white-box setting. Thus, *the comparison with [1] is neither fair nor meaningful.* But we will cite [1] in the future.
>
> Specifically, [1] involves compressing or substituting large models, requiring a deeper understanding, which differs from our current framework. We make the assumption that *the internals of large models are entirely unknown*. While training substitute models through obtaining outputs based on given inputs could be another avenue, as mentioned in our future work, 'Foundation models in ZOOPFL can be enhanced by other ways, e.g., auxiliary models, to serve as a complement to foundation models.' Currently, no one in the field has pursued this direction. Additionally, in our setup, each client has a limited amount of training data, potentially challenging the construction of robust substitute models ('For COVID-19, each client only has about 50 samples.').
>
> [1] Xiao, Guangxuan, Ji Lin, and Song Han. "Offsite-tuning: Transfer learning without full model." arXiv preprint arXiv:2302.04870 (2023).
>
> 3. In terms of model privacy, the privacy leakage of a black-box model is closely related to the number of queries [2], but the authors do not provide theoretical or empirical studies on this.
>
> A good question! However, our paper *does not* have such an issue. Why? The black-box models are in each client, and there is no data exchange between clients and servers, or between clients.
>
> Regarding the privacy protection of the model, it's not within the scope of consideration for our current paper. Our focus lies in exploring how to better leverage a black-box model provided by the model suppliers—enhancing the alignment between local task inputs and outputs with the model.
>
> 4. The experimental section lacks ablation experiments with varying levels of noise added on the transformed data and visualizations of transformed data.
>
> - First, we do have the experiments on varying levels of noise, as illustrated in Figure 6(b).
> - As for visualization, we add such extra experiments in the revised version, as illustrated in Appendix F.5.
>
> We hope your concerns will be resolved and the rating of the paper can be increased accordingly. Thank you!

---

> ### Comment · Reviewer_RdGi · 2023-12-04
>
> After carefully reviewing the responses from other reviewers and the author, I have decided to keep my rating and lower the score for soundness and contribution as the author's response did not address my concerns.
>
> The author highlights the main contribution of this paper as "Scenario Exploration." However, in my opinion, this new scenario may not be practical, and the exploration does not seem to be comprehensive.
>
> Regarding the new scenario, a critical question arises: under what circumstances, with the foundation model available on the local client, do we need to set the model as an access-constrained black-box model? The motivation for introducing this specific "local black-box" scenario remains unclear. Typically, the use of a black-box restriction should be based on specific requirements, such as privacy protection needs or scenarios related to property rights, for instance, when the model serves as a prediction API or proprietary software. These situations necessitate essential related experiments on the number of queries (see black-box visual prompting [1] and black-box adversarial reprogramming [2]).
>
> During the rebuttal phase, the author emphasizes this special "local black-box setting," focusing on the inability to perform backpropagation. If this is the case, there may be no need to emphasize the black-box setting. Instead, the paper could consider the adaptation of MeZO [3] to the Federated Learning (FL) scenario from the perspective of reducing computational and memory costs.
>
> Furthermore, in the response regarding "why is it practical," the conclusion that "their proposed black-box FL is the **only** solution" is derived directly and confidently based on the high computation and communication costs of fine-tuning foundation models locally, which is perplexing and appears unrigorous.
>
> Additionally, I noticed that the author selectively chose a comment from a reviewer, labeled the reviewer as "an emotional reviewer," and posted it on social media. I find this behavior to be inappropriate. Such actions, which disrespect the reviewer in active discussion, are truly disheartening and may diminish the enthusiasm of other reviewers for participating in discussions and collectively improving the paper.
>
> [1] Oh, Changdae, et al. "BlackVIP: Black-Box Visual Prompting for Robust Transfer Learning." Proceedings of the IEEE/CVF Conference on Computer Vision and Pattern Recognition. 2023.
> [2] Tsai, Yun-Yun, Pin-Yu Chen, and Tsung-Yi Ho. "Transfer learning without knowing: Reprogramming black-box machine learning models with scarce data and limited resources." International Conference on Machine Learning. PMLR, 2020.
> [3] Malladi, Sadhika, et al. "Fine-Tuning Language Models with Just Forward Passes." arXiv preprint arXiv:2305.17333 (2023).

---

### Official Review · Reviewer_hxrH · 2023-10-30

**Soundness:** 3 good
**Presentation:** 2 fair
**Contribution:** 3 good
**Rating:** 5
**Confidence:** 4

**Summary:**

This paper proposed ZOOPFL, a zeroth-order optimization system for the black-box local model under a federated learning setup. Instead of directly fine-tune the black-box foundational model, ZOOPFL learns input surgery and semantic re-mapping for black-box large foundation models in federated learning. ZOOPFL aims to adapt inputs to models and project outputs to meaningful semantic space. The experiment shows that ZOOPFL performs better than the ZS setup in both NLP and CV benchmark datasets.

**Strengths:**

1. In the current foundational model era, the black-box foundational model is becoming popular. It is important to propose some ideas to efficiently personalize the foundational model without direct interference with it. Compared to other existing works related to foundational model with FL, the proposed ZOOPFL is the first to achieve federated learning with large black-box models, which is very relative to the current challenges.

2. The paper is well-written and clearly structured. The author selects two different data modalities to validate the soundness of the proposed ZooPFL.

**Weaknesses:**

1. The idea seems very similar to the soft-prompt training [1], which is also working on the input surgery without directly inference the foundational model. What is the benefit of the auto-encoder pre-training in your paper?

2. What are the benefits of personalization? In the experiment part, it mainly focused on the overall accuracy boost compared to ZS, which does not reflect anything regarding to personalization.

3. I suggest the paper should be more clear about the only baseline ZS. I checked several times in the paper, and I could not understand what ZS stands for and why it is a suitable baseline for ZooPFL.


[1]. Wang, Zifeng et al. “Learning to Prompt for Continual Learning.” 2022 IEEE/CVF Conference on Computer Vision and Pattern Recognition (CVPR) (2021): 139-149.

**Questions:**

1. What does ZS stand for in the paper? Does it stand for zero-shot training?

2. I am curious why the author selected the personalized FL as a topic to discuss. Even without step 3, this paper still makes its point regarding how to efficiently use the black-box model under FL setup.

3. I am not very clear why ZooPFL needs Semantic re-mapping. Could the author elaborate more on this?

---

> ### Author Response · Authors · 2023-11-14
> **Response To Reviewer hxrH**
>
> Thanks for your acknowledgment in our paper: *the first to achieve federated learning with large black-box models* and *well-written and clearly structured*! We see that your main concerns are on details of *novelty*, *personalization*, *module functions*, and *concepts*. Now we answer them here.
>
> 1. The idea seems very similar to the soft-prompt training [1], which is also working on the input surgery without directly inference the foundational model. What is the benefit of the auto-encoder pre-training in your paper?
>
> We appreciate your recognition of the significance of [1] and we have cited this work in the revision. Where is the benefits of auto-encoder pre-training?
>
> We use auto-encoders to alleviate the computational complexity associated with zeroth-order optimization. The rationale behind pre-training the auto-encoder stems from the need to adapt it to samples and tasks effectively. The pre-training step, as depicted in Figure 4 of the original manuscript, serves as a crucial aspect of our methodology, as illustrated by the ablation experiments.
>
> In summary, we agree with your assessment and appreciate the opportunity to clarify the unique contributions of our work in addressing the where, how, and why aspects of input manipulation, particularly in the context of personalized federated learning with large black-box models. We will certainly emphasize the distinctions and innovations of our approach in the revised manuscript, giving due credit to the relevant literature, including Reference [1].
>
> [1] Wang, Zifeng et al. “Learning to Prompt for Continual Learning.” 2022 IEEE/CVF Conference on Computer Vision and Pattern Recognition (CVPR) (2021): 139-149.
>
> 2. What are the benefits of personalization? In the experiment part, it mainly focused on the overall accuracy boost compared to ZS, which does not reflect anything regarding to personalization. Even without step 3, this paper still makes its point regarding how to efficiently use the black-box model under FL setup.
>
> Indeed, personalized learning stands out as one of the **most crucial and practical challenges** in FL [2], and our approach naturally tackles this issue. While we do furnish personalized outcomes for each client (as depicted in Figure 3 in the main text and Tables 4-6 in the appendix), constraints posed by page limitations compel us to emphasize average results in our report. It's worth noting that we explicitly stated in the original text, '**Our method achieves the best accuracy in most clients.**'
>
> Without Step 3 or implementing Step 3 via FedAVG, our method seamlessly adapts to the common federated learning (FL) setting. However, the incorporation of Step 3 not only **enhances semantic matching** but also leads to **superior overall outcomes and personalized effects**.
>
> [2] Jian Xu, Xinyi Tong, and Shao-Lun Huang. Personalized federated learning with feature alignment and classifier collaboration. In The Eleventh International Conference on Learning Representations, 2023
>
> 3. What does ZS stand for in the paper? Does it stand for zero-shot training?
>
> The meaning of ZS is introduced in the paper: ZS denotes "zero-shot pre-trained models", i.e., zero-shot inference using the pre-trained models. And with all due respect, the term "zero-shot training" in your question is incorrect since we cannot "zero-shot" train a model if there is no sample available.
>
> 4. I am not very clear why ZooPFL needs Semantic re-mapping. Could the author elaborate more on this?
>
> The benefit of semantic remapping is introduced in the main paper. Here we explain it more. Short answer: the model outputs do not match the target task, so a re-mapping is needed.
>
> Long answer: As highlighted in our main text, "Some popular language models such as BERT and GPT-2 in Huggingface utilize a random projection between extracted features and logits," and "This step endeavors to re-map logits into meaningful semantic spaces with a simple linear projection." The intentional inclusion of a random mapping followed by a specific semantic mapping is justified in practical terms. Moreover, when meeting data not covered by foundation models, foundation models can be able to extract meaningful features but be unable to map it to expected target labels. Semantic re-mapping eliminates these gaps. The ablation experiments further substantiate the significance of this component.
>
> - - -
>
> We hope your concerns will be resolved and the rating of the paper can be increased accordingly. Thank you!

---

> > ### Comment · Reviewer_hxrH · 2023-11-17
> >
> > Thank you for the rebuttal, which clarifies most of my concerns.
> >
> > However, I agree with the Reviewer MDQp that most of the black-box models are hosted in the cloud, and transmitting data to the cloud indeed violates the privacy promise of FL. As a result, your application scenario is now limited to the locally-inference models, which means ZOOPFL is another novel way to efficiently fine-tune the large models without modifying the weights. If my understanding is correct, then PROMPTFL could also be used to fine-tune the black-box models, as it only appends a prompt learner in front of the large foundational model. Please let me know if my understanding is not correct.

---

> ### Author Response · Authors · 2023-11-17
>
> Thank you for the response. But we are upset that reviewer still cannot understand our situation.
>
> First of all, we are not dealing with cloud-hosted models but local ones on each client. The privacy concern makes this setting even more primary. Your understanding about this is correct. But this is not a minor setting.
>
> Second, promptfl requires gradient to tune the prompt, which is not possible in the blackbox setting. As we discussed extensively in the related work part, prompt tuning is not possible.
>
> We are glad that you read other review. But please also read other response to get a better understanding especially our general response.

---

> > ### Comment · Reviewer_MDQp · 2023-11-18
> >
> > I guess the question (at least to me) is: since zeroth order optimization is a known algorithm, can it be applied to prompt tuning as well? As long as a zeroth order algorithm is used, there is no need to compute the gradients. How would applying zeroth order optimization to prompt tuning compare to applying it in your current approach?

---

> > > ### Author Response · Authors · 2023-11-18
> > >
> > > Yes, indeed, ZOO can be utilized for prompt tuning. Please note that our framework is **not limited** to using an Autoencoder for input modification; we can also employ prompt tuning. Additionally, we can replace the use of CGE with MeZO for improved efficiency, exchange the linear layer of semantic remapping with more complex structures such as attention for enhanced results.
> > >
> > > However, these details are not the focal point. The key emphasis is that our work represents the **initial exploration** in this direction, as mentioned in the paper. Our contributions lie in Scenario Exploration, ZOOPFL Framework, Theoretical Support, and Empirical Validation. Future work can delve into other aspects, such as prompt tuning, for better results.

---

> > > > ### Comment · Reviewer_MDQp · 2023-11-18
> > > >
> > > > Initial explorations should be submitted to workshops. Conference papers (at least in the field of machine learning) should generally be a piece of established work.

---

> > > > > ### Author Response · Authors · 2023-11-18
> > > > >
> > > > > We totally understand that the reviewer wants a solid and comprehensive work, but an initial study does not mean it’s not. In fact, conferences do encourage original and initial studies as long as they benefit the community. Thus, we respectfully disagree with your comments that the initial study should go to workshops.
> > > > >
> > > > > In fact, there are many conference papers claiming they are initial studies, such as [1][2][3][4][5]. You can also interpret our initial exploration as the first work in this direction : )
> > > > >
> > > > > Finally, despite all this, please indicate whether our response addresses your initial concerns in the review comment or not. Thanks!
> > > > >
> > > > > [1] Chien, Eli, Chao Pan, and Olgica Milenkovic. "Efficient model updates for approximate unlearning of graph-structured data." The Eleventh International Conference on Learning Representations. 2022.
> > > > >
> > > > > [2] Yuan, Honglin, et al. "What Do We Mean by Generalization in Federated Learning?." International Conference on Learning Representations. 2021.
> > > > >
> > > > > [3] Bosselut, Antoine, Ronan Le Bras, and Yejin Choi. "Dynamic neuro-symbolic knowledge graph construction for zero-shot commonsense question answering." Proceedings of the AAAI conference on Artificial Intelligence. Vol. 35. No. 6. 2021.
> > > > >
> > > > > [4] Gao, Zhihan, et al. "Earthformer: Exploring space-time transformers for earth system forecasting." Advances in Neural Information Processing Systems 35 (2022): 25390-25403.
> > > > >
> > > > > [5] Wang, Yidong, et al. "Exploring Vision-Language Models for Imbalanced Learning." International Journal of Computer Vision (IJCV) 2023.

---

> > > > > > ### Comment · Reviewer_MDQp · 2023-11-18
> > > > > >
> > > > > > I see that an author of this paper has posted on social media that I’m an emotional reviewer and would like to reject this paper due to this comment. That’s not my point. My main point in the comment about “initial exploration” is that all full papers should be rigorous with the claims being sufficiently validated, either theoretically or empirically. The paper should make a contribution that is new and interesting to the research community. It is fine to have limitations, but they should be acknowledged. As I mentioned in the comments at the top, this paper in its current form lacks rigor and there is clearly space for improvement. I’m happy to raise the score if such improvements can be made.
> > > > > >
> > > > > > I shall also say that I am disappointed that the authors seem to be trying to “bypass” the comment about prompt tuning in this thread, by saying that their framework is general enough and can include “everything”, but they do not consider it due to “initial exploration”. In my opinion, this is neither a meaningful argument nor a rigorous way of thinking. Instead, I would expect the authors to explain **why** they use their current method in this study and do not consider alternatives like prompt tuning.
> > > > > >
> > > > > > I am also a bit disappointed that the authors use emotional words like “upset” in their response. While I can understand the frustration, such words are not helpful for a meaningful discussion. To be honest, I rarely (if ever) see such kind of wording or attitude in other discussions.
> > > > > >
> > > > > > Please do not post emotional comments on social media. As a reviewer, my main goal is to make a proper evaluation for every paper to my best effort, for the fairness of both the authors and everyone in the community. I also try to help the authors improve their paper whenever possible. However, if the author says in their social media post themself that I should reject the paper, I am not sure what I should do, but I will still try my best to make a professional judgement and not be biased by such non-technical factors.

---

> > > > > > > ### Author Response · Authors · 2023-11-19
> > > > > > >
> > > > > > > Thank you very much for your professional feedback. Based on your professional feedback, we have made further revisions to the article. Additionally, to address your concerns, we are providing the results of PromptFL+ZOO here. While PromptFL performs better than ZS, it still lags behind our approach due to the absence of considerations for semantic remapping and personalization. Once again, we appreciate your feedback.
> > > > > > >
> > > > > > > | Backbones    | ABLERT | BERT  | DeBERTa | GPT2  |
> > > > > > > |--------------|--------|-------|---------|-------|
> > > > > > > | ZS           | 60.76  | 54.82 | 62.11   | 41.26 |
> > > > > > > | PromptFL+ZOO | 61.22  | 62.55 | 65.73   | 44.19 |
> > > > > > > | Ours         | 68.91  | 68.54 | 68.69   | 71.79 |

---

> > > > > > > > ### Comment · Reviewer_MDQp · 2023-11-21
> > > > > > > >
> > > > > > > > Thanks for the update. It may be good to add this result to the paper as well, unless the authors think otherwise.

---

> > > > > > > > > ### Author Response · Authors · 2023-11-21
> > > > > > > > >
> > > > > > > > > Thank you very much for your feedback. PromptFL is currently an Arxiv paper and we think there might be further updates. Therefore, the results vs PromptFL above is only partial and incomplete, which is used to resolve your concerns in this short period of time. Plus, we are planning to include the comprehensive results by including more datasets and networks as in the main paper, which would take a lot of time.

---

> > > > > > > > > > ### Comment · Reviewer_MDQp · 2023-11-21
> > > > > > > > > >
> > > > > > > > > > Sounds good to me. No problem.

---

### Official Review · Reviewer_1Crz · 2023-10-31

**Soundness:** 3 good
**Presentation:** 3 good
**Contribution:** 3 good
**Rating:** 5
**Confidence:** 3

**Summary:**

The authors designed a method that treats foundation models as black boxes. The idea is to use zeoth-order optimisation to  make sure that fine-tuning can be done efficiently on-device (e.g., to train through federated learning)

**Strengths:**

- Being able to fine-tune Foundation Models (FM) in a privacy-preserving way is an important problem
- Overall this paper helps us understand the scenario of incorporating FM in FL settings.
- The use of zeroth-order optimization, input surgery and semantic re-mapping are interesting contributions here

**Weaknesses:**

Some areas to improve:

- While the idea of using FM as black box is interesting, there might be some privacy implications. It is unclear if the input to the FM reveals any information about the private input that is used both for training and during inference. I assume that the FM --being a black box and too big to be hosted on-device-- is run externally). As a result, this method might limit the ability of FL to offer privacy-preserving training.
In other words,  If we assume that the "black box" in figure 2.b runs externally, what are the privacy implications wrt to its input and output crossing the device boundary. If it runs on-device then what are the assumptions wrt to its size and the fact that it is a black box.

- The authors assume that the FM is a black box. With more and more FM being open-sourced, it would be great if the authors can further motivate their approach and what might be the main advantages of incorporating a black box.

- The evaluation is mostly done on rather simple benchmarks. I was wondering if the proposed approach (to train just parts of the model) would carry enough capacity to tackle larger tasks. Maybe some discussion or even evaluation on a more complex task would be great.

- The paper might benefit from some understanding of the memory footprint and computation complexity of this method. Overall, the main target of this method is to make FM training possible with FL (on-device). As a result, we should have a good understanding on the memory/computation overhead.

**Questions:**

See above

---

> ### Author Response · Authors · 2023-11-14
> **Response To Reviewer 1Crz**
>
> Thanks for your acknowledgment of the paper: *an important problem*, *helps us understand the scenario of incorporating FM in FL settings*, and *interesting contributions*! We see that your main concerns are on details of *privacy implication*, *more FM*, *larger task*, and *the memory footprint and computation complexity*. Now we answer them here.
>
> 1. If we assume that the "black box" in figure 2.b runs externally, what are the privacy implications wrt to its input and output crossing the device boundary. If it runs on-device then what are the assumptions wrt to its size and the fact that it is a black box.
>
> - First, we currently do not assume cross-device data exchange following existing work on PFL. So, the privacy issue does not exist.
> - Second, for on-device training of our model, the sizes of models and data highly depend on the local computation hardware, but not our algorithm. By design, our algorithm supports all kinds of models and datasets.
>
> 2. The authors assume that the FM is a black box. With more and more FM being open-sourced, it would be great if the authors can further motivate their approach and what might be the main advantages of incorporating a black box.
>
> Please refer to the general response for the detailed answer.
>
> Short answer: even if you can have full access to the model locally, it is still expensive to actually train them. "Black-box" does not equal to close-source, even open-source models are expensive to train on ordinary devices.
>
> 3. The evaluation is mostly done on rather simple benchmarks. I was wondering if the proposed approach (to train just parts of the model) would carry enough capacity to tackle larger tasks. Maybe some discussion or even evaluation on a more complex task would be great.
>
> In actuality, our study encompassed extensive experimentation across **eight** datasets that include **two modalities**: COVID-19, APTOS, Terra100, Terra46, SST2, COLA, Financial-phrasebank (Financial), and Flipkart. The results from these experiments underline the superiority of the framework proposed in our study.
>
> Please be aware that the tasks we have selected are currently **not adequately addressed or poorly performed** by large models. Our aim is to align these large models with specific tasks, **rather than undergoing retraining**. Consequently, we do not require an abundance of data or a larger dataset.
>
> 4. The paper might benefit from some understanding of the memory footprint and computation complexity of this method. Overall, the main target of this method is to make FM training possible with FL (on-device). As a result, we should have a good understanding on the memory/computation overhead.
>
> The answer is simple: we only introduce *negligible* memory and computation burden to the foundation models.
>
> As depicted in Figure 4(d) of the original paper, it's evident that the parameters adjusted by our method are significantly smaller in scale compared to the large foundation models. This aspect reinforces the notion that our approach hones in on a smaller subset, effectively **tailoring the model's adjustments**, rather than necessitating the manipulation or direct handling of the entire foundation model. Our emphasis remains on the utilization of these large models as protected tools within federated learning, ensuring their functionality while mitigating the challenges associated with their training costs.
>
> - - -
>
> We hope your concerns will be resolved and the rating of the paper can be increased accordingly. Thank you!

---

### Official Review · Reviewer_MDQp · 2023-11-06

**Soundness:** 3 good
**Presentation:** 3 good
**Contribution:** 3 good
**Rating:** 6
**Confidence:** 4

**Summary:**

The paper presents a method for personalized federated learning (FL) while relying on the existence of foundation models at clients. The main idea is to train some additional components (auto-encoder and semantic re-mapping) that are applied before or after the foundation model. Zeroth-order optimization has been applied due to the assumption that the foundation model cannot be accessed for purposes other than inference. Experimental results confirm the advantage of the proposed method compared to some baselines.

**Strengths:**

- The consideration of foundation models in FL is an important research direction.

**Weaknesses:**

- The paper assumes that foundation models are located at FL clients, but the clients cannot perform back-propagation on these models. It is not clear in what practical scenario such an assumption would hold. It is worth noting that most large language models (LLMs) nowadays are hosted in the cloud. Obviously, transmitting data to the cloud, even for inference, violates the privacy promise provided by FL. It seems that the authors of this paper try to overcome this privacy violation by assuming that the foundation model is hosted on each client. However, this has several issues. First, many types of LLMs are not feasible to run on mobile devices, which means that the proposed approach may only be possible in the case of cross-silo FL but not cross-device FL. Second, and more importantly, if the foundation model is hosted at the client, it is unclear why gradients cannot be computed, since each client has full access to its model in this case.
- Overall, the proposed approach is a combination of several known techniques, including zeroth-order optimization, so the novelty seems limited.
- The method requires additional components to be added to an existing foundation model, which appears to be a patch instead of a long-term solution. These additional components will cause additional computational overhead, which has not been studied in the paper.

**Questions:**

My questions are related to the weaknesses mentioned above, which are summarized as follows:
- In what practical scenario would a FL client host a foundation model, but does not have full access to it?
- What are the key technical challenges and novel solution in this work?
- What is the additional computational overhead of the additional components (auto-encoder and semantic re-mapping) in the proposed method, when the full combined model is used for inference? It would be helpful to measure and compare the inference time with and without these additional components on a real device.

---

> ### Author Response · Authors · 2023-11-14
> **Response To Reviewer MDQp**
>
> Thanks for your acknowledgment in our *research direction*! We see that your main concerns are on details of *practical scenario*, *novelty*, and *computational overhead*. Now we answer them here.
>
> 1. In what practical scenario would a FL client host a foundation model, but does not have full access to it?
>
> Please refer to the general response for a better understanding of our problem setting.
>
> Short answer: *The understanding of the word "black-box" could be broad.* Black-box does not only deal with the case where one has no full access to the model; even if you have the model access, it is still extremely expensive to run model updates (Backpropagation) on the local devices given the increasing larger sizes of the models. As an example used in the paper, 'training GPT-2-small (Radford et al., 2019) requires at least two A100 GPUs for 16 hours, a resource unavailable to many'. We are a solution of using inference for BP to save computation and communication costs.
>
> 2. What are the key technical challenges and novel solution in this work?
>
> Key technical challenges:
> - efficiently update the foundation models locally
> - distribution divergence in different clients
>
> Our innovations to deal with them:
> - A first pioneering exploration of the problem where models become too large to update, while FL clients still want to preserve privacy by not sharing data with the central server.
> - A new framework to handle the problem using zeroth order optimization.
> - We offer theoretical analysis along with extensive experiments to show the effectiveness of our work.
>
> 3. What is the additional computational overhead of the additional components (auto-encoder and semantic re-mapping) in the proposed method, when the full combined model is used for inference?
>
> Simple: the additional computation costs are *negligible* compared to the huge cost of the foundation models. Quantitatively speaking, as depicted in Figure 4(d), the parameter volume of the head-tail adjustment module is **several orders of magnitude smaller** compared to the foundation model.
>
> If you think these responses address your concerns, please consider increasing your score. Thank you!

---

### Author Response · Authors · 2023-11-14
**General response on our problem setting**

We thank all reviewers for your efforts in reviewing this paper. However, we found that most of the reviewers **significantly misunderstood** our problem setting, leading to negative comments. We would like to explain here.

**What is our setting?**

We view the large foundation models as black-box that can only provide the outputs according to inputs. We have **no access** to any internal information on the foundation models, which means, **no backpropagation** is allowed for updating. Each client preserves the same foundation model **locally**. Most importantly, we do **not consider** the storage of large models and the additional costs associated with inference. We aim to utilize large black-box foundation models for better personalized federated learning.

**Why is it practical?**

To make the best of large foundation models in federated learning, one must:
- either fine-tune or adapt the models in their own data,
- or perform federated learning on the cloud.

Therefore, it is easy to see the value of our work, since:
- fine-tuning or adapting locally is extremely expensive even if we have many open-source foundation models.
	- Why? Because fine-tuning on client side requires high computation and communication costs. By "client side", we do not mean just mobile phone devices, but any organization (e.g., a hospital) could be a client to be a part of the FL cycle.
- federated learning on the cloud is not the ideal solution if you care about privacy.
	- Why? One cannot trust the cloud providers by uploading all the training data to the cloud. So, the best practice is to perform computation *locally*.

Combining the above situations. i.e., updating models locally with low cost, one can conclude that our proposed black-box FL is the only solution. Specifically, note that "black-box" does not only mean we do not have model access; it is a more broad technique for model update when you cannot perform local BP due to large model sizes.

- - -

We certainly hope that reviewers can have a better understanding of our problem setting based on the above explanation.

---

> ### Comment · Reviewer_MDQp · 2023-11-18
>
> Thanks for the response. The authors claim that their proposed black-box FL has "low cost". However, I do not see anywhere in the experiments that measures this cost compared to baselines.
>
> Furthermore, the writing of the paper needs substantial improvement. For example:
> - The experiments use "ZS" as a baseline, but it is not clear what ZS stands for (I couldn't find it from the paper).
> - In Theorem 1, I cannot find the definitions of the quantities $\Delta_\mathbf{u}$ and $\Delta_\mathbf{v}$.
> - In Algorithm 1 (in Appendix C), $s_i$ and $r_i$ are part of the output, but the algorithm does not seem to include any step for computing $s_i$ and $r_i$. How are they computed?
>
> Unfortunately, the more carefully I read the paper, the more I find the paper to be lacking scientific rigor. Even cleaning up all the definitions and other details requires a thorough and careful revision, besides other issues. It is clearly not ready for publication in its current form.

---

> ### Author Response · Authors · 2023-11-18
>
> Thank you very much for your feedback. We would like to briefly address your questions here, and we plan to upload a revised version in the next couple of days:
>
> 1. Experiments on costs.
>
> In Figure 4(c), we mentioned the number of parameters, stating that fewer parameters imply lower cost under similar conditions.
>
> 2. The meaning of "ZS".
>
> ZS means zero-shot and we have added the corresponding description.
>
> 3. The definitions of the quantities.
>
> The definitions are in Eq. 41 on page 18 of the original paper.
>
> 4. How are s and r computed.
>
> 's' and 'r' represent the respective networks, and we use 'q' to denote the parameters to be calculated. 'r' can be optimized directly using SGD/ADAM, and we will make it clear in the revised version.
>
> We greatly appreciate your detailed feedback, which has contributed to the improvement of our paper. If you have any additional comments or suggestions, please feel free to share them. Your insights are valued, and we look forward to further refining our work based on your input.

---

> > ### Comment · Reviewer_MDQp · 2023-11-18
> >
> > Thanks. Regarding “In Figure 4(c), we mentioned the number of parameters, stating that fewer parameters imply lower cost under similar conditions,” the number of parameters cannot capture the difference in cost between forward propagation only and forward+backward propagation, which is the key difference between zeroth order optimization and existing first-order based approaches. Some other metric, such as FLOPs, memory consumption, or running time, would be needed for a meaningful comparison between the two types of approaches.

---

> > > ### Author Response · Authors · 2023-11-18
> > >
> > > Thanks for the comments. We will run extra experiments later to answer your questions onf flops, memory, and running time.

---

> > > > ### Comment · Reviewer_MDQp · 2023-11-18
> > > >
> > > > Thanks. If the authors can make the edits as discussed, to improve the clarity and rigor of the paper, I would be happy to raise my score.
> > > >
> > > > In general, the main paper should be self-contained with all the variables properly defined. Having variables appearing in the main theorem defined in the appendix is not appropriate in my opinion, since reviewers and other readers are not expected to read the appendix in order to understand the main conclusions in the paper.

---

> > > > > ### Author Response · Authors · 2023-11-18
> > > > >
> > > > > Thank you very much for your professional and constructive reviews. We are diligently working on the necessary revisions and aim to release an updated version within the next two days. Once again, we appreciate your valuable feedback.

---

### Author Response · Authors · 2023-11-19
**Reminder of the revised version upload**

Dear Reviewers,

Thank you very much for the valuable feedback from each reviewer. Based on the suggestions, we have thoroughly revised our manuscript, including but not limited to, adding references, adjusting the article structure, refining descriptions, and incorporating new experiments and analyses. We hope our revised version effectively addresses your concerns. Once again, we appreciate the input from everyone.

Thanks a lot!

---

### Author Response · Authors · 2023-11-21
**Reminder of discussion**

Dear Reviewers,

We appreciate your time and feedback greatly. Particularly, through interactions with the reviewer MDQp, our paper has improved significantly. As the rebuttal period is coming to a close, do you have any further questions or issues?

Thanks a lot!

---

> ### Comment · Reviewer_MDQp · 2023-11-21
>
> Thanks for the updates. For Table 3 containing the new results, could you explain why your method consumes less GPU memory although it includes additional processing? Do you use more CPU instead of GPU?
>
> In addition, as mentioned in my [earlier comment](https://openreview.net/forum?id=U2ZIgcrg7Z&noteId=e61wNpnwNu), it would be nice to compare the difference in cost between forward propagation only and forward+backward propagation, which is the key difference between zeroth order optimization and existing first-order based approaches. The current results do not seem to include this comparison. In other words, it would be nice to include some empirical evidence showing the advantage of zeroth-order optimization in terms of computational efficiency.

---

> > ### Author Response · Authors · 2023-11-21
> >
> > Thank you very much for your feedback. Except for the GPU-related costs, which record the consumption during training, all other costs reflect the inference phase. It is reasonable that our approach consumes less GPU than forward and backward since we do not need to compute and store the gradients of foundational models. We do not claim that ZOO can improve computational efficiency in this paper. It is just one available solution to handle the black box situation and make finetuning large models in clients possible. In fact, it is not that efficient as pointed out by several existing papers [1][2]. The efficiency of ZOO is out of the scope of this paper.
> >
> > [1] Malladi, Sadhika, et al. "Fine-Tuning Language Models with Just Forward Passes." NeurIPS 2023.
> >
> > [2] Zhang, Yimeng, et al. "How to Robustify Black-Box ML Models? A Zeroth-Order Optimization Perspective." International Conference on Learning Representations. 2022.

---

> > > ### Comment · Reviewer_MDQp · 2023-11-21
> > >
> > > Thanks for the explanation. Could you perhaps list the cost of training and cost of inference in separate tables (or subtables), and add a bit more explanation to each of them? I think even the fact that ZOO saves GPU memory during training is a nice characteristic of it, which is worth highlighting.
> > >
> > > I'll raise my score to 6.

---

> > > > ### Author Response · Authors · 2023-11-21
> > > >
> > > > Thank you for your timely feedback and recognition of our work. We will promptly incorporate your professional insights, make the necessary revisions, and upload a new version. Once again, thanks a lot.

---

### Author Response · Authors · 2023-11-22
**General Response**

Dear Reviewers,

Thank you very much to all the reviewers for your valuable reviews. We understand that most concerns are about the problem setting of our research, which may influence the assessment of our contributions. We have provided clarifications on these concerns and kindly ask for your reconsideration.

We appreciate the feedback from Reviewer MDQp and look forward to hearing from Reviewer 1Crz, Reviewer hxrH, and Reviewer RdGi.

Thank you all!

---

### Meta-Review · Area_Chair_cL5s · 2023-12-07

**Metareview:**

Summary:
The paper presents a method for personalized federated learning (FL) while relying on the existence of foundation models for clients. The main idea is to train some additional components (auto-encoder and semantic re-mapping) applied before or after the foundation model. Zeroth-order optimization has been applied due to the assumption that the foundation model cannot be accessed for purposes other than inference. Experimental results confirm the advantage of the proposed method compared to some baselines.

Strengths:
+ The consideration of foundation models in FL is an important research direction.
+ Overall this paper helps us understand the scenario of incorporating FM in FL settings.
+ The use of zeroth-order optimization, input surgery, and semantic re-mapping are interesting contributions here.

Weaknesses:
- The paper assumes that foundation models are located at FL clients, but the clients cannot perform back-propagation on these models. It is not clear in what practical scenario such an assumption would hold.
- Overall, the proposed approach is a combination of several known techniques, including zeroth-order optimization, so the novelty seems limited.
- The method requires additional components to be added to an existing foundation model, which appears to be a patch instead of a long-term solution.
- While the idea of using FM as a black box is interesting, there might be some privacy implications.
- The authors assume that the FM is a black box. With more and more FM being open-sourced, it would be great if the authors could further motivate their approach and what might be the main advantages of incorporating a black box.
- The evaluation is mostly done on rather simple benchmarks.
- The paper might benefit from some understanding of the memory footprint and computation complexity of this method.

**Justification For Why Not Higher Score:**

For the reasons above

**Justification For Why Not Lower Score:**

N/A

---

### Decision · Program_Chairs · 2024-01-16

Reject